# Global Warming Potential Estimates for the $C_1$-$C_3$ Hydrochlorofluorocarbons (HCFCs) Included in the Kigali Amendment to the Montreal Protocol

Dimitrios K. Papanastasiou[1,2], Allison Beltrone[3], Paul Marshall[3], and James B. Burkholder[1]

[1]Earth System Research Laboratory, Chemical Sciences Division, National Oceanic and Atmospheric Administration, 325 Broadway, Boulder, CO 80305, USA.

[2]Cooperative Institute for Research in Environmental Sciences, University of Colorado, Boulder, CO 80309 USA.

[3]Department of Chemistry, University of North Texas, P.O. Box 305070, Denton, Texas 76203-5070.

*Correspondence to*: James Burkholder (James.B.Burkholder@noaa.gov)

**Abstract.** Hydrochlorofluorocarbons (HCFCs) are ozone depleting substances and potent greenhouse gases that are controlled under the Montreal Protocol. However, the majority of the 274 HCFCs included in Annex C of the protocol do not have reported global warming potentials (GWPs) that are used to guide the phase-out of HCFCs and the future phase-down of hydrofluorocarbons (HFCs). In this study, GWPs for all $C_1$–$C_3$ HCFCs included in Annex C are reported based on estimated atmospheric lifetimes and theoretical methods used to calculate infrared absorption spectra. Atmospheric lifetimes were estimated from a structure activity relationship (SAR) for OH radical reactivity and estimated $O(^1D)$ reactivity and UV photolysis loss processes. The $C_1$–$C_3$ HCFCs display a wide range of lifetimes (0.3 to 62 years) and GWPs (5 to 5,330, 100-year time horizon) dependent on their molecular structure and H-atom content of the individual HCFC. The results from this study provide estimated policy relevant GWP metrics for the HCFCs included in the Montreal Protocol in the absence of experimentally derived metrics.

## 1 Introduction

Hydrochlorofluorocarbons (HCFCs) are ozone depleting substances (ODSs), the production and use of which are controlled under the Montreal Protocol on Substances that Deplete the Ozone Layer (1987). HCFCs have been used as substitutes for chlorofluorocarbons (CFCs) in various commercial and residential applications, e.g. foam blowing, and refrigerator and air conditioning systems. In addition to being ODSs, HCFCs are also potent greenhouse gases (WMO, 2014).

With the adoption of the Kigali Amendment (2016) to the Montreal Protocol by the Twenty-Eighth Meeting of the Parties to the Montreal Protocol, parties agreed to the phase-down of hydrofluorocarbons (HFCs), substances that are not ozone depleting but are climate forcing agents. As in the case of HCFCs, the HFC production and consumption control measures comprise reduction steps from established baselines (see UN Environment OzonAction Fact Sheet

(UN, 2017)), which are different for developed and developing countries with an exemption for countries with high ambient temperature. Since HFCs are greenhouse gases, baselines and reduction steps are expressed in $CO_2$ equivalents. The amended Protocol controls eighteen HFCs as listed in Annex F of the Protocol.

Although the phase-down steps stipulated in the Kigali Amendment concern only HFCs, the baselines for the reductions are derived through formulae involving both HCFCs and HFCs production and consumption because HFCs are intended to be substitute compounds for HCFCs. This necessitates knowledge of the global warming potentials (GWPs), a policy relevant metric representing the climate impact of a compound relative to $CO_2$, of all HCFCs involved in the baseline formulae. However, in the amended Protocol, GWPs are available for only 8 HCFCs (HCFCs-21, -22, -123, -124, -141b, -142b, -225ca, and HCFC-225cb) out of the total of 274 HCFCs included in Annex C (274 is the sum of all $C_1$–$C_3$ HCFC isomers). Of the 274 HCFCs, only 15 have experimental kinetic and/or infrared absorption spectrum measurements used to determine their GWPs. The majority of the HCFCs listed in Annex C are not currently in use, but the intent of the Protocol was for a comprehensive coverage of possible candidates for future commercial use and possible emission to the atmosphere. For molecules with no GWP available, a provision is included in the Protocol stating that a default value of zero applies until such a value can be included by means of adjustments to the Protocol. Having policy-relevant metrics for these compounds will help guide and inform future policy decisions.

The objective of the present work is to provide a comprehensive evaluation of the atmospheric lifetimes, ozone depletion potentials (ODPs), which represents the ozone depleting impact of a compound relative to a reference compound (see WMO (2014) and references within), GWPs, and global temperature change potentials (GTPs), another policy relevant metric representing the climate impact of a compound relative to $CO_2$, for the HCFCs listed in Annex C of the amended Protocol. The HCFCs that have experimentally measured OH rate coefficients, the predominant atmospheric loss process for HCFCs, and infrared absorption spectra were used as a training dataset to establish the reliability of the methods used to estimate the metrics for the other HCFCs. The training dataset compounds and reference data are listed in Table 1. In the following section, brief descriptions of the methods used to determine the HCFC atmospheric lifetime and ODP are given. Next, the theoretical methods used to calculate the infrared spectra of the HCFCs are described. The infrared spectra are then combined with our estimated global atmospheric lifetimes to estimate the lifetime and stratospheric temperature adjusted radiative efficiency (RE), GWP, and GTP metrics (see IPCC (2013) and WMO (2014)). In the Results and Discussion section, a general overview of the obtained metrics is provided, while the details and results for each of the individual HCFCs are provided in the Supporting Information (SI).

**Table 1.** Summary of hydrochlorofluorocarbon (HCFC) parameters in the training dataset *

| Common Name | Molecular Formula | $k_{OH}$(298 K) (10$^{-14}$ cm$^3$ molecule$^{-1}$ s$^{-1}$) [1] | Atmospheric Lifetime (years) | Stratospheric Lifetime (years) | Infrared Absorption Spectrum Source | Radiative Efficiency (RE) (W m$^{-2}$ ppb$^{-1}$) | Global Warming Potential (GWP) 100-yr time horizon |
|---|---|---|---|---|---|---|---|
| HCFC–21 | CHFCl$_2$ | 3.0 | 1.7 | ~35 | Sihra et al. (2001) | 0.15 | 148 |
| HCFC–22 | CHF$_2$Cl | 0.49 | 11.9 | 161 | Sihra et al. (2001) | 0.21 | 1760 |
| HCFC-31 | CH$_2$FCl | 4.1 | 1.2 | ~35 | – | – | – |
| HCFC–122 | CHCl$_2$CClF$_2$ | 5.1 | – | – | Orkin et al. (2003) | 0.17 | 59 |
| HCFC–122a | CHClFCCl$_2$F | 1.6 | – | – | Orkin et al. (2003) | 0.21 | 258 |
| HCFC–123 | CHCl$_2$CF$_3$ | 3.6 | 1.3 | 36 | Sihra et al. (2001) | 0.15 | 79 |
| HCFC–123a | CHClFCClF$_2$ | 1.3 | 4.0 | ~65 | – | 0.23 | 370 |
| HCFC–124 | CHClFCF$_3$ | 0.90 | 5.9 | 111 | Sihra et al. (2001) | 0.20 | 527 |
| HCFC–124a | CHF$_2$CClF$_2$ | – | ~9.2 | ~120 | Sharpe et al. (2004) | – | – |
| HCFC–132 | CHClFCHClF | – | – | – | Sharpe et al. (2004) | – | – |
| HCFC–132a | CHCl$_2$CHF$_2$ | – | – | – | Sharpe et al. (2004) | – | – |
| HCFC–132b | CHCl$_2$CHF$_2$ | 1.7 | – | – | – | – | – |
| HCFC–132c | CH$_2$FCCl$_2$F | 1.23 | – | – | Orkin et al. (2003) | 0.17 | 338 |
| HCFC–133a | CH$_2$ClCF$_3$ | 1.1 [2] | 4.45 [2] | 103 [2] | Sharpe et al. (2004) Etminan et al.(2014) McGillen et al.(2015) | 0.16 [2] | 370 [2] |
| HCFC–141b | CH$_3$CCl$_2$F | 0.58 | 9.4 | 72.3 | Sihra et al. (2001) Sharpe et al. (2004) | 0.16 | 782 |
| HCFC–142b | CH$_3$CClF$_2$ | 0.34 | 18 | 212 | Sihra et al. (2001) | 0.19 | 1980 |
| HCFC–225ca | CHCl$_2$CF$_2$CF$_3$ | 2.5 | 1.9 | 44 | Sihra et al. (2001) | 0.22 | 127 |
| HCFC–225cb | CHClFCF$_2$CClF$_2$ | 0.89 | 5.9 | 101 | Sihra et al. (2001) | 0.29 | 525 |
| HCFC–234fb | CCl$_2$FCH$_2$CF$_3$ | 0.080 | ~45 | ~85 | – | – | – |
| HCFC–243cc | CH$_3$CF$_2$CFCl$_2$ | 0.24 | 19.5 | ~70 | – | – | – |

\* Lifetimes, RE, and GWP values taken from WMO ozone assessment (WMO, 2014) unless noted otherwise. Where multiple sources for infrared spectra are available, the spectra reported from the NOAA laboratory (McGillen et al., 2015) and the PNNL database (Sharpe et al., 2004) were used in the analysis.
[1] Rate coefficients taken from NASA evaluation (Burkholder et al., 2015) unless noted otherwise.
[2] Rate coefficient and metrics taken from McGillen et al. (2015) with RE lifetime adjusted and a factor of +1.1 for stratospheric temperature correction applied.

## 2 Methods

### 2.1 Atmospheric Lifetimes

The global atmospheric lifetime ($\tau_{atm}$) is defined as:

$$\frac{1}{\tau_{atm}} = \frac{1}{\tau_{OH}} + \frac{1}{\tau_{O(^1D)}} + \frac{1}{\tau_{h\nu}}$$

where $\tau_{OH}$, $\tau_{O(^1D)}$, and $\tau_{h\nu}$ are the global lifetimes with respect to OH and O($^1$D) reactive loss and UV photolysis, respectively. Other reactive and deposition loss processes for HCFCs are expected to be negligible and not considered in this study. $\tau_{atm}$ is also often defined in terms of its loss within the troposphere ($\tau_{Trop}$), stratosphere ($\tau_{Strat}$), and mesosphere ($\tau_{Meso}$) as:

$$\frac{1}{\tau_{atm}} = \frac{1}{\tau_{Trop}} + \frac{1}{\tau_{Strat}} + \frac{1}{\tau_{Meso}}$$

where for example:

$$\frac{1}{\tau_{Strat}} = \frac{1}{\tau_{Strat}^{OH}} + \frac{1}{\tau_{Strat}^{O(^1D)}} + \frac{1}{\tau_{Strat}^{h\nu}}$$

For the HCFCs considered in this study, mesospheric loss processes are negligible and not considered further. The atmospheric loss processes for the HCFCs considered in this study have not been determined experimentally, while $\tau_{Trop}$ is predominately determined by the HCFC reactivity with the OH radical. In this work, $\tau_{Trop}^{OH}$ was estimated using the CH$_3$CCl$_3$ (MCF) relative method (WMO, 2014) where:

$$\tau_{Trop}^{OH} = \tau_{OH}^{HCFC} = \frac{k_{MCF}(272\,K)}{k_{HCFC}(272\,K)}\,\tau_{OH}^{MCF}$$

with the MCF recommended rate coefficient, $k_{MCF}$(272 K) = 6.14 × 10$^{-15}$ cm$^3$ molecule$^{-1}$ s$^{-1}$ (Burkholder et al., 2015), and tropospheric lifetime, 6.1 years (WMO, 2014).

In the absence of experimental OH reaction rate coefficients, a structure activity relationship (SAR) was used to estimate OH reaction rate coefficients. The SAR of Kwok and Atkinson (1995) and DeMore (1996) were compared with the rate coefficients for the 15 HCFCs (training dataset) for which experimental kinetic measurements are available (Burkholder et al., 2015). The DeMore SAR clearly performed better for these halocarbons and was used in this study. Figure 1 shows the agreement between the experimental 298 K rate coefficient data and the SAR predicted values. For the determination of $k_{HCFC}$(272 K) an E/R value of 1400 K was used in the Arrhenius expression, $k$(T) = A exp(-1400/T), which is a representative value for the HCFC reactions included in Burkholder et al. (2015). On the basis of the training dataset calculations, we estimate the uncertainty in the SAR 298 K rate coefficients on average to be ~30%. The uncertainty at 272 K will, in some cases, be greater due to our assumption that $E$/R = 1400 K for the unknown reaction rate coefficients. A ~50% uncertainty spread encompasses nearly all the training dataset values at 272 K, see Figure 1. Therefore, we estimate a 50% uncertainty in $k$(272 K) for the HCFCs with unknown rate coefficients.

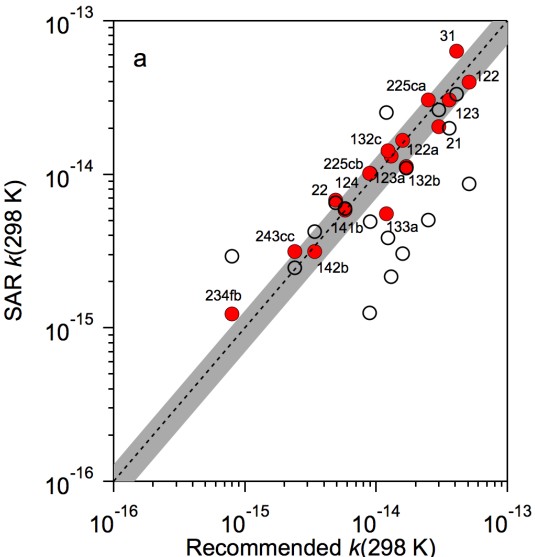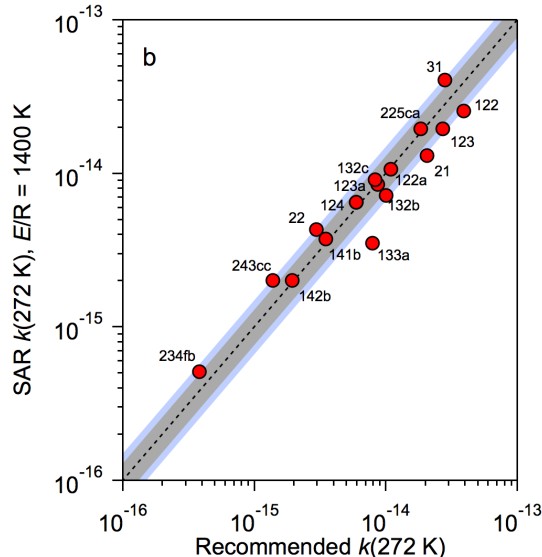

**Figure 1:** Comparison of structure activity relationship (SAR) OH rate coefficients for the training dataset (Table 1) with rate coefficients recommended in Burkholder et al. (2015). (a) Rate coefficients at 298 K using the SAR of DeMore (1996) (solid symbols) and Kwok and Atkinson (1995) (open symbols). The dashed line is the 1:1 correlation and the shaded region is the ±30% spread around the 1:1 line. (b) Rate coefficients at 272 K using the SAR of DeMore (1996) (solid symbols) with an $E/R = 1400$ K. The dashed line is the 1:1 correlation and the gray shaded region is the ±30% spread and the blue shaded region is the ±50% spread around the 1:1 line.

$\tau_{Strat}$ for the HCFCs is determined by a combination of OH and O($^1$D) reactive loss, and UV photolysis. Presently, there is not a simple means to determine stratospheric lifetimes without the use of atmospheric models. Here, we have estimated stratospheric OH loss lifetimes, $\tau_{Strat}^{OH}$, following a methodology similar to that used in the WMO (2014) ozone assessment, where results from 2-D atmospheric model calculations are used to establish a correlation between tropospheric and stratospheric lifetimes. We have used the lifetimes taken from the SPARC (Ko et al., 2013) lifetime report for 3 HCFCs and 8 HFCs to establish a lifetime correlation, which is shown in Figure S1 in the SI. The stratospheric loss via the OH reaction accounts for ~≤5% of the total OH loss process for >95% of the HCFCs. Therefore, this method of accounting for stratospheric loss leads to only a minor uncertainty in the calculated global lifetime.

In most cases, O($^1$D) reaction and UV photolysis are minor contributors to the global loss of a HCFC. In the absence of experimental data, O($^1$D) rate coefficients were estimated using the reactivity trends reported in Baasandorj et al. (2013). $\tau_{O(1D)}$ for the HCFCs were based on a comparison with similarly reactive compounds included in the SPARC (Ko et al., 2013) lifetime report. As shown later, the O($^1$D) reaction is a minor loss process, <1%, for nearly all the HCFCs included in this study and, therefore, the estimation method used is not critical as this loss process is a minor contributor to the global lifetime. $\tau_{hv}$ was estimated based on the molecular Cl-atom content and its distribution within

the molecule as follows: each isolated Cl-atom (450 years), each $CCl_2$ group (80 years), each $CCl_3$ group (50 years), with a minimum photolysis lifetime of 50 years (Ko et al., 2013). UV photolysis is a minor loss process, with the exception of a few long-lived highly-chlorinated HCFC isomers where photolysis accounts for at most 15% of the global loss.

A minimum stratospheric lifetime of 20 years was applied to approximately account for transport limited stratospheric lifetimes.

## 2.2 Ozone Depletion Potentials (ODPs)

Semi-empirical ODPs were calculated using the formula:

$$\text{ODP}_{HCFC} = \frac{n_{Cl}}{3} \frac{f_{HCFC}}{f_{CFC-11}} \frac{M_{CFC-11}}{M_{HCFC}} \frac{\tau_{HCFC}}{\tau_{CFC-11}}$$

where $n_{Cl}$ is the number of Cl-atoms in the HCFC, M is the molecular weight, $f$ is the molecules fractional release factor (FRF), which denotes the fraction of the halocarbon injected into the stratosphere that has been dissociated (Solomon and Albritton, 1992), and $\tau$ is the global atmospheric lifetime. The fractional release factor and global lifetime for CFC-11 were taken from the WMO (2014) ozone assessment report to be 0.47 and 52 years, respectively. Note that a new method to calculate FRF has been suggested by Ostermoeller et al. (2017a,b), which has been applied

by Leedham Elvdige et al. (2018) and Engel et al. (2018). Overall, there is good agreement between the new method and the empirical parameterization applied in this work. The fractional release factors for the majority of the HCFCs included in this study have not been reported. The WMO report included 3 year age of air FRFs derived from model studies and field observations for 20 ozone depleting substances (WMO, 2014). In the absence of recommended FRF values, we derived an empirical FRF vs stratospheric lifetime relationship, shown in Figure 2, for the compounds with

reported FRFs and the 2-D model stratospheric lifetimes reported in the SPARC (Ko et al., 2013) lifetime report. Table S1 provides the values presented in Figure 2. A fit to the data yielded FRF = 0.06 + 0.875 × exp(-0.0144 × $\tau_{Strat}$), which was used in our calculations.

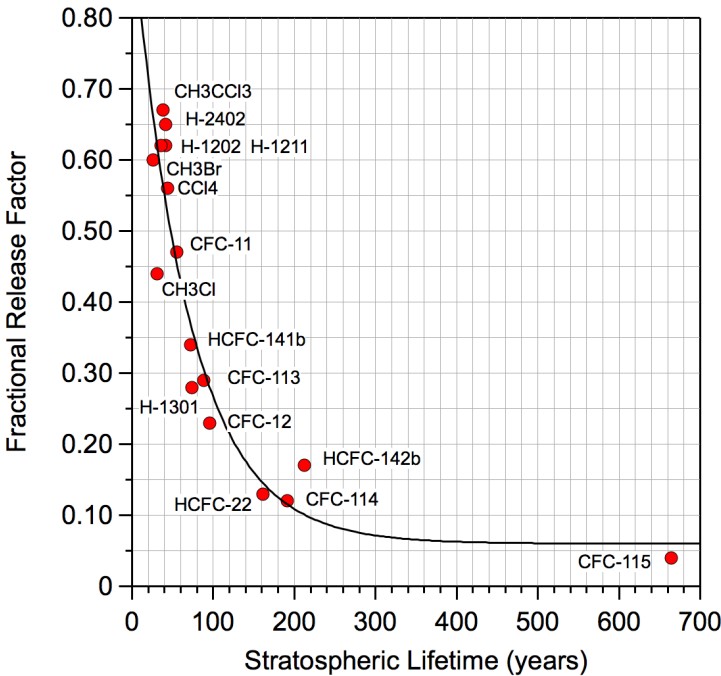

**Figure 2:** Empirical correlation of fractional release factor (FRF) versus stratospheric lifetime, $\tau_{Strat}$. Stratospheric lifetimes were taken from 2-D model results given in the SPARC (Ko et al., 2013) lifetime report. The FRFs were taken from WMO assessment (WMO, 2014). The solid line is a fit to the data: $FRF = 0.06 + 0.875\exp(-0.01444 \times \tau_{Strat})$.

## 2.3 Theoretical Calculations

Information about molecular vibrational frequencies, central to the interpretation of infrared spectra, thermodynamics, and many other aspects of chemistry, became amenable to computational determination in the early 1980s. It was recognized that computed harmonic frequencies derived via the second derivative of energy as a function of atomic position were systematically higher than observed fundamentals and scale factors were introduced (Hout et al., 1982; Pople et al., 1981). For Hartree-Fock frequencies these were typically ~0.9 and accounted both for the influence of anharmonicity and deficiencies in the underlying quantum calculations. Frequencies based on methods incorporating electron correlation such as CCSD, CCSD(T) or certain functionals within density functional theory (DFT) often perform well for harmonic frequencies and are scaled by ~0.95 to match fundamental vibrational modes. Such scaling has been updated as more methods appear (Alecu et al., 2010; Scott and Radom, 1996). Rather less information is available concerning the evaluation of absorption intensities for fundamental modes. Within the same harmonic approximation, implemented in popular quantum codes, the intensity is proportional to the square of the derivative of the dipole moment with respect to position. Halls and Schlegel evaluated QCISD results against experiment and their plot indicates deviations of up to around ±20% and then used QCISD as a benchmark to evaluate a range of functionals (Halls and Schlegel, 1998). For B3LYP, they found differences from QCISD of around 10%. More recently, tests of the B3LYP functional found good performance for frequency and intensity (Jiménez-Hoyos et al., 2008; Katsyuba et

al., 2013). Some prior work where similar methods have been applied to the infrared absorption for molecules of atmospheric interest include studies of fluoromethanes (Blowers and Hollingshead, 2009), unsaturated hydrofluorocarbons (Papadimitriou and Burkholder, 2016; Papadimitriou et al., 2008b), perfluorocarbons (Bravo et al., 2010), chloromethanes (Wallington et al., 2016), $SO_2F_2$ (Papadimitriou et al., 2008a), permethylsiloxanes (Bernard et al., 2017), and large survey studies such as by Kazakov et al. (Kazakov et al., 2012) and Betowski et al. (Betowski et al., 2015) to name a few.

Halls and Schlegel noted that real spectra may exhibit the influences of resonances, intensity sharing, and large-amplitude anharmonic modes. These can be partially accounted for in an analysis based on higher derivatives of the energy and the dipole moment, performed for instance within the framework of second-order vibrational perturbation theory (Barone, 2005). Advantages include treatment of resonances among vibrational levels and incorporation of overtones and combination bands. Examples of applications to molecules containing C-H and C-F bonds indicate excellent accord with experiment for band position and intensity, (Carnimeo et al., 2013) but for $CH_2ClF$ the intensity in the region involving C-Cl stretching nevertheless exhibits intensity errors of ~10% (Charmet et al., 2013).

The large number of molecules considered in this work and the associated geometry optimizations, ~1500 optimizations, required that a cost-effective methodology with reasonable accuracy such as density functional theory (DFT) methods be used. Geometry optimization and vibrational frequencies for all $C_1$-$C_3$ HCFCs were carried at the B3LYP/6-31G(2df,p) level using the Gaussian 09 software suite (Frisch et al., 2016). Similar approaches have been used in earlier studies for other classes of molecules with good results, see Hodnebrog et al. (2013) and references cited within. The calculations presented in this work included only the [35]Cl isotope because the large number of possible isotopic substitution permutations made the calculation of all combinations prohibitive. In principle, substitution of [35]Cl by [37]Cl in a heavy molecule would lower the frequency of the C-Cl stretch by ~3%. The level of theory was evaluated based on comparison with available experimental HCFC infrared spectra, see Table 1. Note that our calculations and data available in the NIST quantum chemistry database (2016) obtained using a more costly triple-$\zeta$ basis set (aug-cc-pVTZ) showed only minor differences in the calculated frequencies, <1%, and band strengths, <10%, for the molecules in the training dataset.

The majority of the HCFCs have multiple low-energy conformers that have unique infrared absorption spectra. Although only the most stable conformer has been used in most previous theoretical studies, including the individual conformers provides a more realistic representation of the HCFCs infrared spectrum and is expected to improve the accuracy of the calculated radiative efficiency as discussed below. We are not aware of prior studies of infrared spectra of HCFC conformers, but there have been prior theoretical studies of the conformers of other classes of molecule, such as for validation of observed infrared spectra used to deduce relative energies of carbonyl conformations (Lindenmaier et al., 2017) and comparison with measured infrared intensities for linear alkanes (Williams et al., 2013). The different errors and their trends for the intensities of C-H stretching and HCH bending modes indicate that a simple scaling approach, so successful for frequencies, will not work for intensities. In this work, we have included all conformers within 2 kcal mol$^{-1}$ of the lowest energy conformer. This limit accounts for

>98% of the population distribution at 298 K, in most cases. For each HCFC, a relaxed scan was performed to detect all possible conformations. For the $C_2$ compounds, 3 staggered conformations were examined by rotating the C-C torsional angle by 120º. For the $C_3$ compounds, 9 possible conformations were calculated by rotating the two torsional angles by 120º. Each stable conformer was then fully optimized at the B3LYP/6-31G(2df,p) level followed by a frequency calculation. Conformer populations were calculated for a 298 K Boltzmann's distribution using the relative energies (including a zero-point correction) from the calculations. Including stable conformers resulted in overlapping vibrational bands and, therefore, more congested spectra which is consistent with the observed spectra for HCFCs. A number of the HCFCs have stereoisomers. Although, the stereoisomers have identical infrared absorption spectra, they were accounted for in the population distribution. Note that for a molecule with a single asymmetrical carbon (a molecule containing a carbon with 4 different groups attached), e.g. HCFC-121a ($CHClFCCl_3$), a pair of stereoisomers exist for each conformation and, therefore, the contribution of stereoisomers to the total population factors out. The entire dataset contains 126 molecules with a single asymmetric carbon and 32 molecules containing 2 asymmetric carbons.

A comparison of the experimental and calculated infrared spectrum of HCFC-124a ($CHF_2CClF_2$) shown in Figure 3 demonstrates the importance of including conformers in the spectrum calculation. A comparison of experimental and theoretical spectra for all molecules with experimental data is provided in the SI (see Section 5). The calculations found that HCFC-124a has 3 stable conformers at 298 K with the lowest energy conformer having ~50% of the population. The experimental spectrum is characterized by strong absorption features between 1100 and 1500 $cm^{-1}$, which are mostly associated with C-F bond vibrations, and C-Cl vibrational modes below 1000 $cm^{-1}$. The comparison with the experimental spectrum shows that the prominent absorption features at ~825, 1000, and 1250 $cm^{-1}$ originate from the higher energy conformers. The calculated spectrum is in good agreement with the experimentally measured spectrum with band positions and total integrated band strengths agreeing to within ~2%. Note that conformer contributions to an infrared absorption spectrum will be different for different molecules. The impact of including conformers in the radiative efficiency calculations is presented later.

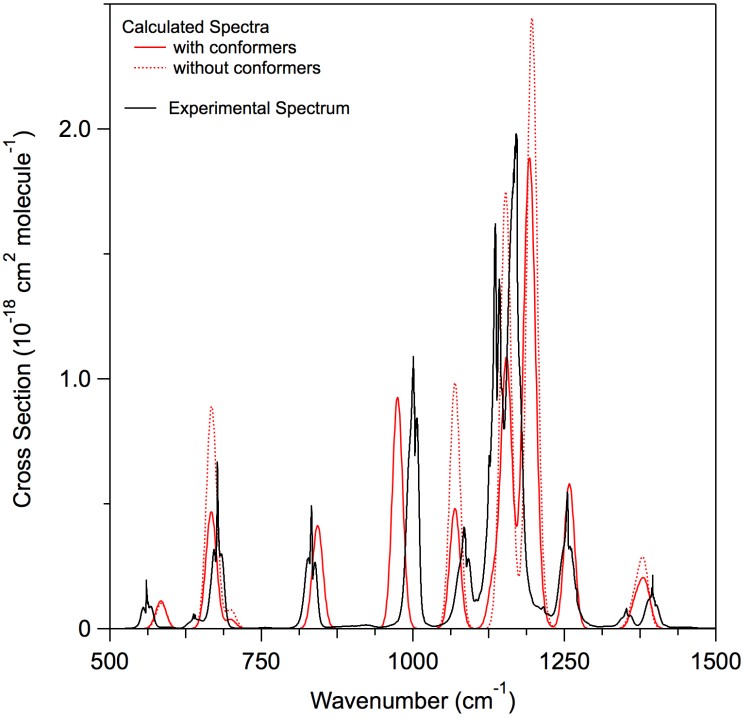

**Figure 3:** Comparison of experimental and calculated infrared absorption spectrum of HCFC-124a (CHF$_2$CClF$_2$). (a) Calculated spectra at B3LYP/6-31G(2df,p) level of theory with (solid) and without (dotted) including stable conformers, and (b) the experimentally measured spectrum (see Table 1 for the source of the experimental spectrum).

Overall, the agreement between experimental and calculated frequencies was good. Figure S2 shows a comparison of experimental vibrational frequencies with the calculated values. There was a systematic overestimation of the calculated vibrational frequencies above 1000 cm$^{-1}$ and an underestimation below 1000 cm$^{-1}$. An empirical frequency correction, which, in part, accounts for anharmonicity and other approximations used in the level of theory employed, was derived from this correlation and applied to all the calculated spectra: $\nu_{corrected} = 53.609 + 0.94429 \times \nu_{calculated}$. Using this correction, frequencies around ~1200 cm$^{-1}$ (C-F bond vibrations) and around 800 cm$^{-1}$ (C-Cl bond vibrations) are shifted by only ~1%. The uncertainty associated with the calculated band positions is estimated to be ~1%. The frequency-corrected spectra were used to derive the metrics reported here.

Figure 4 shows a comparison of calculated and experimental band strengths (integrated between 500 and 2000 cm$^{-1}$) for the training dataset. Overall, the agreement is good for the majority of HCFCs with the calculated band strengths being within 20%, or better, of the experimental values. The calculated band strengths are, however, systematically biased high by ~20%, for band strengths <~$1.1 \times 10^{-16}$ cm$^2$ molecule$^{-1}$ cm$^{-1}$. A comparison of the training dataset experimental and calculated infrared spectra reveals that the bias originates from a band strength overestimation of bands below 1000 cm$^{-1}$ that are primarily associated with C-Cl bonds. The bias is greatest for molecules containing more than one Cl atom on the same carbon, e.g. CHFCl$_2$ (HCFC-21), CH$_3$CCl$_2$F (HCFC-141b) and CH$_2$FCCl$_2$F (HCFC-132c). In fact the intensities of C-Cl stretches are a long-known problem for calculation (Halls and Schlegel,

1998).  Scaling the overall spectrum strength to account for such biases has been applied to decrease the deviation between experimental and theoretical values in an earlier theoretical study by Betowski et al. (2015).  However, since the bias is primarily for the bands associated with C-Cl bonds, a scaling of the entire band strength would not be appropriate nor an accurate representation of the experimental spectrum.  The spectra reported here do not include a band strength correction, as the prediction of which bands are overestimated is too uncertain without knowledge of the experimental spectrum.  Although it is difficult to estimate the uncertainty for the theoretical calculations, an estimated ~20% band strength uncertainty includes nearly all the training dataset values and encompasses the possible systematic bias observed for certain vibrational bands.

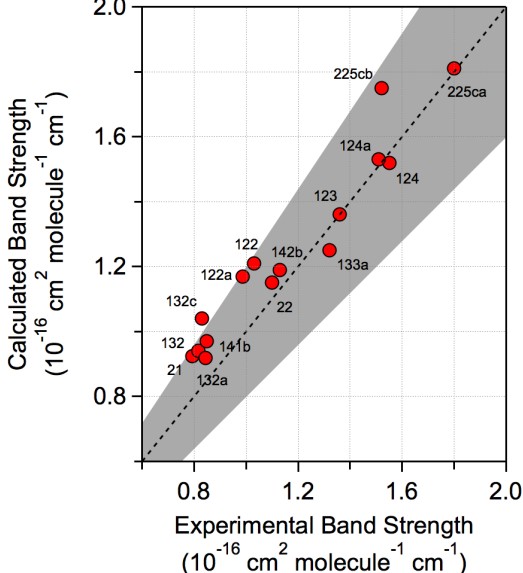

**Figure 4**: Comparison of experimental and calculated infrared band strengths over the 500–2000 cm$^{-1}$ region for the HCFC training dataset (see Table 1 for the source of the experimental spectra).  The dashed line is the 1:1 correlation.  The shaded region represents a 20% spread around the 1:1 line.

## 2.4  Radiative Efficiency

Radiative efficiencies (REs) were calculated using the 298 K infrared absorption spectra calculated theoretically in this work and the Earth's irradiance parameterization given in Hodnebrog et al. (2013).  The calculated spectra were broadened using a Gaussian broadening function with a FWHM (full width at half maximum) of 20 cm$^{-1}$, which reproduces the training dataset spectra reasonably well and provides a more realistic representation of the spectrum and overlap with Earth's irradiance profile. Note that the Gaussian broadening function may not necessarily be an accurate representation of the actual vibrational band shape.  Previous theoretical studies of greenhouse gases have applied band broadening as part of their analysis (see Hodnebrog et al. (2013) and references within), although the necessity of broadening has generally not been treated quantitatively.  In our work, the calculated bands were

broadened to obtain better agreement with available experimental HCFC spectra, which are assumed to be representative of the spectra of the unknown HCFCs, and, in principle, more reliable radiative efficiencies. Figure 5 shows the difference in retrieved REs with and without band broadening. The differences are molecule dependent, but are less than 10% for nearly all the HCFCs. Although the differences are relatively small the use of a realistic broadening function reduces the uncertainty in the RE calculation and should be applied. A comparison of the experimentally derived REs and the calculated values for the training dataset is given in Figure S3.

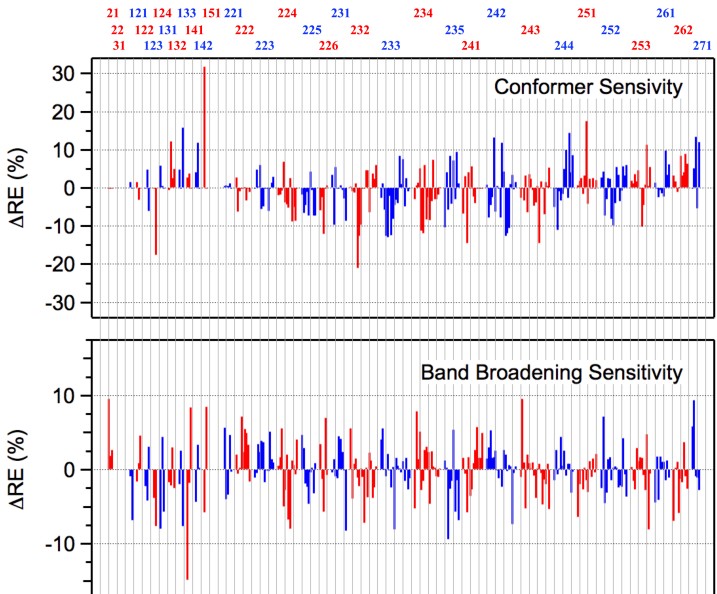

**Figure 5:** Sensitivity of the calculated HCFC radiative efficiencies in this study to the broadening of the calculated infrared absorption bands, as described in the text, (Lower Panel) and the inclusion of higher energy conformers (Upper Panel). ΔRE values are relative to the full analysis that includes broadened spectra and all conformers within 2 kcal mole$^{-1}$ of lowest energy conformer.

As illustrated earlier for HCFC-124a, Figure 3, stable HCFC conformers can make a significant contribution to its infrared absorption spectrum. Figure 5 shows the impact of including the conformer population on the calculated RE for each of the HCFCs included in this study. Overall, including conformers increases or decreases the calculated RE by 10%, or less, in most cases. However, there are some HCFCs where a difference of 20%, or more, is observed, e.g. HCFC-124a, HCFC-151, and HCFC-232ba. In conclusion, including the contribution from populated conformers improves the accuracy of the calculated RE values and decreases potential systematic errors in the theoretically predicted RE values.

The strongest HCFC vibrational bands are due to C-F stretches, 1000-1200 cm$^{-1}$, which strongly overlap the "atmospheric window" region. The molecular geometry of the HCFC determines the exact vibrational band frequencies, i.e., HCFCs and their isomers have unique infrared absorption spectra and REs. Note that the calculated

infrared spectra in this work include vibrational bands below 500 cm$^{-1}$, which is usually the lower limit for experimental infrared absorption spectra measurements. The contribution of vibrational bands in this region to the RE is quantified in our calculations and is usually minor, i.e., <1%. The Earth's irradiance profile, HCFC infrared absorption spectra, and HCFC radiative efficiency spectra for each HCFC included in this study are included in the Supplementary Material (see Section 5).

Lifetime-adjusted REs were calculated using the CFC-11 emission scenario "S" shaped parameterization given in Hodnebrog et al. (2013), which is intended to account for non-uniform mixing of the HCFC in the atmosphere. The adjustment is greatest for short-lived molecules. A +10% correction was applied to all molecules to account for the stratospheric temperature correction (see IPCC (2013) supplementary material section 8.SM.13.4 for the origin of this factor). Well-mixed and lifetime-adjusted RE values are included in the Supporting Information datasheets (see Section 5).

### 2.5 Global Warming and Global Temperature change Potentials

Global warming potentials on the 20- and 100-year time horizons (T) were calculated relative to $CO_2$ using the formulation given in IPCC (2013):

$$\text{GWP(T)} = \frac{\text{RE}\, \tau\, [1 - \exp{(-\text{T}/\tau)}]}{\text{M}_{\text{HCFC}}\, \text{Int}\, \text{RF}_{\text{CO}_2}(\text{T})}$$

where IntRF$_{CO2}$(T) is the integrated radiative forcing of $CO_2$ and M$_{HCFC}$ is the HCFC molecular weight. The RE used in the calculation was lifetime-adjusted with a stratospheric temperature correction applied. The global lifetimes were estimated as described in Section 2.1. The $CO_2$ denominator is consistent with the GWP values reported in the WMO (2014) and IPCC (2013) assessments corresponding to a $CO_2$ abundance of 391 ppm. Therefore, the values reported in this work can be compared directly to values reported in the WMO and IPCC assessments. A comparison of our training dataset values is given in Figure 6, where the majority of the GWPs agree to within 15%. HCFCs 21, 22, 122, and 123 have larger differences, due primarily to discrepancies between the estimated and literature OH rate coefficients. Our GWP results can be scaled to the 2016 $CO_2$ abundance of 403 ppm (NOAA, 2017) by multiplying by 1.03, which accounts for a decrease in the $CO_2$ radiative efficiency (see Myhre et al. (1998) and Joos et al. (2013)).

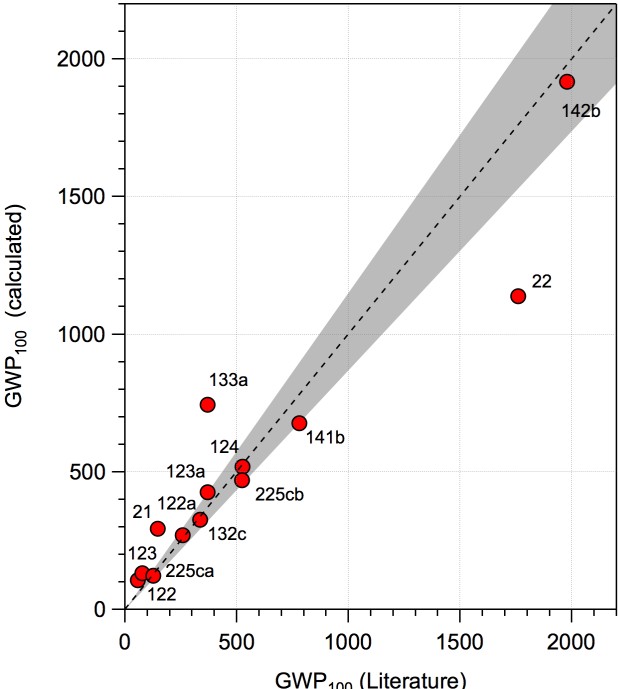

**Figure 6.** Comparison of 100-year time horizon GWP values reported in the WMO assessment (WMO, 2014) and McGillen et al. (2015) for 133a (with lifetime-adjustment and stratospheric temperature correction applied) and the values calculated in this study. The dashed line represents the 1:1 correlation and the shaded area is a 15% spread around the 1:1 line.

Global temperature change potentials were calculated for the 20-, 50-, and 100-year time horizons using the parameterizations given in the IPCC (2013) supplementary material section 8.SM.11.2.

### 3 Results and Discussion

Figure 7 provides a comprehensive graphical summary of the lifetime, ODP, lifetime and stratospheric temperature adjusted RE, GWP, and GTP results obtained in this study and the values that are based on experimental data (in black) where available. The metric values for the individual compounds are available in Table S2 and the individual datasheets in the Supporting Information. A detailed summary of the theoretical results is also included in the datasheets for the individual compounds.

It is clear that the metrics for the $C_1$-$C_3$ HCFCs possess a significant range of values with a dependence on the H-atom content as well as the isomeric form for a given chemical formula. In general, an increase in the HCFC H-atom content leads to a shorter atmospheric lifetime, e.g. the lifetimes for the HCFC-226 compounds (1 H-atom) are greater than most other HCFCs. However, the HCFC reactivity also depends on the distribution of hydrogen, chlorine, and fluorine within the molecule, i.e., the isomeric form and lifetimes for isomers can vary significantly. For example, the lifetime of HCFC-225ca ($CHCl_2CF_2CF_3$) is 1.9 years, while that of HCFC-225da ($CClF_2CHClCF_3$) is 16.3 years. The highest reactivity HCFCs are short-lived compounds with lifetimes as low as ~0.3 years. The lowest reactivity HCFCs have lifetimes as long as 60 years (HCFC-235fa, $CClF_2CH_2CF_3$).

The trends in the HCFC ODPs follow that of the lifetimes with an additional factor to account for the chlorine content of the HCFC. Overall many of the HCFCs have significant ODPs with 33 HCFCs having values greater than 0.1 and 78 greater than 0.05.

5     In addition to HCFC isomers having different reactivity (lifetimes), each isomer also has a unique infrared absorption spectrum and, thus, a unique RE. The HCFC REs range from a low of ~0.03 to a high of ~0.35 W $m^{-2}$ $ppb^{-1}$. The HCFCs with the highest H-atom content have lower REs, in general, although there are exceptions as shown in Figure 6. As expected, many of the HCFCs are potent greenhouse gases. The GWPs and GTPs also show a strong isomer dependence, e.g. the GWPs on the 100-year time horizon for the 9 HCFC-225 isomers differ by a factor of ~12. The lowest HCFC GWPs in this study are ~10 and the greatest value is ~5400 for HCFC-235fa.

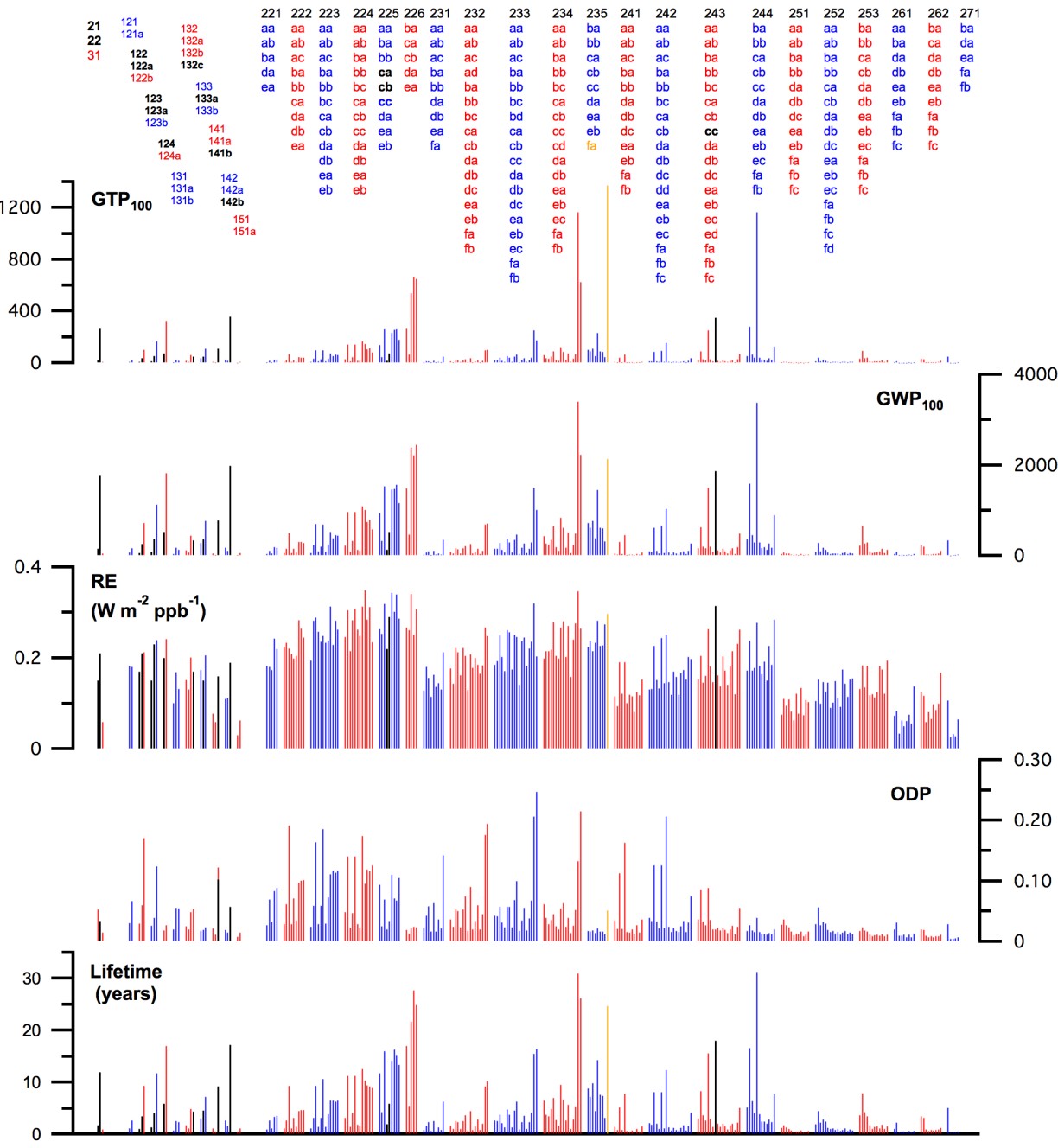

**Figure 7:** Summary of the results obtained in this study for $C_1$-$C_3$ HCFCs (red and blue) and the values for which experimentally derived metrics are available (black). The lifetime, GWP, and GTP values for HCFC-235fa ($CClF_2CH_2CF_3$) (gold) have been multiplied by 0.4 to improve the overall graphical clarity.

### 3.1 Metric Uncertainty

The training calculations have been used to estimate the uncertainties in our atmospheric lifetime estimates and infrared absorption spectra and how these uncertainties propagate through to the key ODP, RE, GWP, and GTP metrics. It is not possible to assign a single uncertainty for all HCFCs for each metric due to their dependence on the individual properties of the HCFCs. To provide a general perspective for the reliability of the metrics reported in this study, we limit our discussion to the average behavior.

The predominant atmospheric loss process for HCFCs was shown to be reaction with the OH radical, while UV photolysis in the stratosphere was found to be a non-negligible loss process for HCFCs with long lifetimes and significant Cl content. The DeMore (1996) SAR predicts the training dataset OH rate coefficients at 298 K to within 25% on average, which directly translates into a 25% uncertainty in the HCFC tropospheric lifetime. A conservative uncertainty estimate in the predicted OH rate coefficients at 272 K would be ~50%, see Figure 1. Including an estimated ~40% uncertainty for the stratospheric UV photolysis and $O(^1D)$ reactive loss processes increases the global lifetime uncertainty by only ~2%.

The semi-empirical ODP uncertainty is directly proportional to the global lifetime uncertainty with an additional factor to account for the uncertainty in the fractional release factor (FRF). For HCFCs with total lifetimes less than 2 years, the total ODP uncertainty is estimated to be 35%, for a 25% uncertainty in the global lifetime. For longer lived HCFCs, the ODP uncertainty is greater, 50% or more.

The theoretically calculated REs for the training dataset agreed to within 10% with the values derived using the experimentally measured spectra, even though our estimated band strength uncertainty is 20%. This method of RE determination is, therefore, expected to provide good estimates of REs in the absence of experimentally based determinations. The accuracy of the lifetime-adjusted RE values depends primarily on the uncertainty in the lifetime, calculated infrared band strengths, and to a lesser degree on the calculated band positions. We estimated the uncertainty in the calculated band strengths to be <20%, although not all vibrational bands are expected to have the same level of uncertainty. The uncertainty in the calculated vibrational band centers primarily impacts the RE through changes in the overlap with the irradiance profile. This sensitivity was estimated by artificially shifting the calculated spectrum in the RE calculation of several representative HCFCs by the estimated band center uncertainty of 1%. The band center uncertainty was found to make less than a 5% contribution to the total RE uncertainty. Note that molecules with strong absorption features near the large $CO_2$ and $O_3$ dips in the Earth's irradiance profile would have a greater sensitivity to shifts in the spectrum. In such cases, direct laboratory studies would be invaluable in the determination of the molecules radiative properties. The lifetime correction to the RE introduces an additional uncertainty that is dependent on the lifetime of the HCFC and its uncertainty. For compounds with a lifetime of 0.2 to 1 year, i.e., on the steep portion of the lifetime correction profile given in Hodnebrog et al. (2013), an additional ~25% uncertainty is introduced, while for longer-lived HCFCs the lifetime adjustment uncertainty is smaller. A lifetime-adjusted RE uncertainty of ~30% is estimated for the majority of the HCFCs included in this study assuming a 25% uncertainty in the global lifetime.

The overall uncertainty in the GWP and GTP metrics depends on the lifetime and RE uncertainties, with a different dependence on different time horizons. Compounds with lifetimes of less than 1 year have propagated uncertainties of ~55% on average. As the lifetime increases the uncertainty decreases to ~30% on average, or less. The greater uncertainty values for the shorter lived HCFCs is primarily associated with the uncertainty introduced by the lifetime-

adjusted RE.

As mentioned earlier, there have been a number of previous studies that have applied methods similar to those used in the present study. The most relevant of these studies is that of Betowski et al. (2015) who reported radiative efficiencies for a large number of the $C_1$-$C_3$ HCFCs included in this study. Although they report REs for 178 of the 274 HCFCs included in our work there are significant differences between their REs and those reported here. Figure

S4 shows a comparison of the RE values calculated here with those reported in Betowski et al. for the HCFCs common to both studies. The RE values from Betowski et al. are systematically lower than the ones reported here by ~29% on average. A similar systematic underestimation is observed when the Betowski et al. RE values are compared with the available HCFC experimental data used in our training dataset. Betowski et al. used B3LYP/6-31G(d) to calculate the HCFC infrared spectra and applied a band strength correction in their RE calculation. Note that a band strength

correction was not applied in the present study as discussed earlier. In addition, Betowski et al. did not use broadened infrared spectra in their RE calculation and included only the lowest energy conformer. These differences can account for some of the scatter in the correlation shown in Figure S4. The average difference between the reported RE values can only partially be explained by the different methods used here, B3LYP/6-31G(2df,p), and in Betowski et al., B3LYP/6-31G(d), as they produce very similar HCFC infrared spectra, i.e., the band strengths obtained with these

methods agree to within ~10%.

Betwoski et al. used the available HCFC experimental data and data for a large number of compounds from other chemical classes in their training dataset, e.g. perhalocarbons, haloaldehydes, haloketones, and haloalcohols. On the basis of their analysis, a band strength scaling factor of 0.699, for the B3LYP/6-31G(d) method, was derived. However, for the HCFCs this scale factor introduces a systematic error in the band strength analysis. In Figure 4 we

showed that the DFT theoretical methods, without scaling, agree with the available experimental HCFC data to within 20%, or better. Although the HCFC training dataset is relatively small, the band strength scaling factor based on results for other chemical compound classes is most likely not appropriate and introduces a systematic bias for the calculated RE values. Therefore, the infrared spectra reported in the present work and used to derive REs and GWPs were not scaled.

**4 Summary**

In this study, policy-relevant metrics have been provided for $C_1$-$C_3$ HCFC compounds, many of which were not available at the time of the adoption of the Kigali amendment. Table 2 summarizes the results from this study in the condensed format used in Annex C of the amended Protocol where the range of metrics are reported for each HCFC chemical formula. Metrics for the individual HCFCs are given in Table S2 and the data sheets for each of the HCFCs

that contain the explicit kinetic parameters and theoretical results obtained in this work.

**Table 2.** The Annex C HCFC table provided in the Kigali amendment to the Montreal Protocol, where the range of 100-year time horizon global warming potentials (GWPs) obtained in this work for various HCFC isomers all with the chemical formula listed in the first column is given in *italics* *

| Group | Substance | Number of isomers | Ozone Depletion Potential*** | 100-Year Global Warming Potential**** |
|---|---|---|---|---|
| *Group I* | | | | |
| $CHFCl_2$ | (HCFC-21)** | 1 | 0.04 | 151 |
| $CHF_2Cl$ | (HCFC-22)** | 1 | 0.055 | 1810 |
| $CH_2FCl$ | (HCFC-31) | 1 | 0.02 | *47* |
| $C_2HFCl_4$ | (HCFC-121) | 2 | 0.01–0.04 | *66–158* |
| $C_2HF_2Cl_3$ | (HCFC-122) | 3 | 0.02–0.08 | *105–713* |
| $C_2HF_3Cl_2$ | (HCFC-123) | 3 | 0.02–0.06 | *130–1125* |
| $C_2HF_3Cl_2$ | (HCFC-123)** | – | 0.02 | 77 |
| $C_2HF_4Cl$ | (HCFC-124) | 2 | 0.02–0.04 | *517–1826* |
| $C_2HF_4Cl$ | (HCFC-124)** | – | 0.022 | 609 |
| $C_2H_2FCl_3$ | (HCFC-131) | 3 | 0.007–0.05 | *31–175* |
| $C_2H_2F_2Cl_2$ | (HCFC-132) | 4 | 0.008–0.05 | *67–441* |
| $C_2H_2F_3Cl$ | (HCFC-133) | 3 | 0.02–0.06 | *273–762* |
| $C_2H_3FCl_2$ | (HCFC-141) | 3 | 0.005–0.07 | *15–676* |
| $C_2H_3FCl_2$ | (HCFC-141b)** | – | 0.11 | 725 |
| $C_2H_3F_2Cl$ | (HCFC-142) | 3 | 0.008–0.07 | *108–1916* |
| $C_2H_3F_2Cl$ | (HCFC-142b)** | – | 0.065 | 2310 |
| $C_2H_4FCl$ | (HCFC-151) | 2 | 0.003–0.005 | *11–54* |
| $C_3HFCl_6$ | (HCFC-221) | 5 | 0.015–0.07 | *38–181* |
| $C_3HF_2Cl_5$ | (HCFC-222) | 9 | 0.01–0.09 | *56–495* |
| $C_3HF_3Cl_4$ | (HCFC-223) | 12 | 0.01–0.08 | *56–693* |
| $C_3HF_4Cl_3$ | (HCFC-224) | 12 | 0.01–0.09 | *83–1090* |
| $C_3HF_5Cl_2$ | (HCFC-225) | 9 | 0.02–0.07 | *122–1562* |
| $C_3HF_5Cl_2$ | (HCFC-225ca)** | – | 0.025 | 122 |
| $C_3HF_5Cl_2$ | (HCFC-225cb)** | – | 0.033 | 595 |
| $C_3HF_6Cl$ | (HCFC-226) | 5 | 0.02–0.10 | *467–2452* |
| $C_3H_2FCl_5$ | (HCFC-231) | 9 | 0.05–0.09 | *17–346* |
| $C_3H_2F_2Cl_4$ | (HCFC-232) | 16 | 0.008–0.10 | *26–713* |
| $C_3H_2F_3Cl_3$ | (HCFC-233) | 18 | 0.007–0.23 | *38–1496* |
| $C_3H_2F_4Cl_2$ | (HCFC-234) | 16 | 0.01–0.28 | *55–3402* |
| $C_3H_2F_5Cl$ | (HCFC-235) | 9 | 0.03–0.52 | *315–5327* |
| $C_3H_3FCl_4$ | (HCFC-241) | 12 | 0.004–0.09 | *10–452* |
| $C_3H_3F_2Cl_3$ | (HCFC-242) | 18 | 0.005–0.13 | *29–1027* |
| $C_3H_3F_3Cl_2$ | (HCFC-243) | 18 | 0.007–0.12 | *34–1498* |
| $C_3H_3F_4Cl$ | (HCFC-244) | 12 | 0.009–0.14 | *124–3369* |
| $C_3H_4FCl_3$ | (HCFC-251) | 12 | 0.001–0.01 | *9–70* |
| $C_3H_4F_2Cl_2$ | (HCFC-252) | 16 | 0.005–0.04 | *24–275* |
| $C_3H_4F_3Cl$ | (HCFC-253) | 12 | 0.003–0.03 | *57–665* |
| $C_3H_5FCl_2$ | (HCFC-261) | 9 | 0.002–0.02 | *7–84* |
| $C_3H_5F_2Cl$ | (HCFC-262) | 9 | 0.002–0.02 | *28–227* |
| $C_3H_6FCl$ | (HCFC-271) | 5 | 0.001–0.03 | *5–338* |

\* Typos for HCFC 123 and 124 GWPs entries are corrected here.

\*\* Identifies the most commercially viable substances.

\*\*\* The ODPs listed are from the Montreal Protocol, while ODPs derived in This Work for the individual HCFCs are available in the Supporting Material, Table S2.

\*\*\*\* Range of values from This Work obtained for the HCFC isomers are given in italics.

We have shown that HCFC isomers have significantly different lifetimes, ODPs, and radiative metrics. Of particular interest are the HCFCs with current significant production and emissions to the atmosphere. Of all the HCFCs listed in Annex C of the amended Protocol, HCFCs -121(2), -122(3), -133(3), 141(3), -142(3), and -225(9) are of primary interest (the values in parenthesis are the number of isomers for that chemical formula). Of these 23 compounds, experimentally based metrics are included in the Kigali amendment only for HCFCs -141b, -142b, -225ca, and -225cb. Therefore, the present work provides policy-relevant information for the other HCFCs.

Although this work has provided a comprehensive set of estimated metrics for the $C_1$-$C_3$ HCFCs that presently do not have experimental data, careful direct fundamental laboratory studies of an intended HCFC would better define the critical atmospheric loss processes (reaction and UV photolysis) used to evaluate atmospheric lifetimes. Laboratory measurements of infrared spectra would also provide specific quantitative results to be used in the determination of the RE, GWP, and GTP metrics. It is anticipated that laboratory measurements could yield uncertainties in the reactive and photolysis loss processes of ~10% and the infrared spectrum of ~5%, or better, which are significantly less than the 25% and 20% average estimated uncertainties obtained with the methods used in this work. Therefore, laboratory studies would potentially yield more accurate metrics. Note that the absolute uncertainty in the ODP, RE, GWP, and GTP metrics would also include a consideration of the uncertainties associated with lifetime determination methods and the Earth's irradiance profile approximation used to derive RE values, as well as the uncertainty in $CO_2$ radiative forcing, which were not considered in this work.

## 5 Data availability

Figures and tables including the master summary table of metrics for all HCFCs is provided in the supporting material. Data sheets for the individual HCFCs that contain the derived atmospheric lifetimes, ODP, RE, GWP, and GTP metrics and graphs and figures and tables of the theoretical calculation results are available at URL https://www.esrl.noaa.gov/csd/groups/csd5/datasets/.

## 6 Competing interests

The authors have no financial conflicts.

## 7 Acknowledgments and Data

This work was supported in part by NOAA Climate Program Office Atmospheric Chemistry, Carbon Cycle, and Climate Program and NASA's Atmospheric Composition Program. The authors acknowledge helpful discussion with Sophia Mylona of the United Nations Environment Programme and David Fahey. The authors acknowledge the NOAA Research and Development High Performance Computing Program (http://rdhpcs.noaa.gov) and the University of North Texas Chemistry cluster purchased with support from the NSF Grant CHE-1531468 for providing computing and storage resources that contributed to the research results reported within this paper.

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
