# Peer review of "Global Warming Potential Estimates for the C1-C3 Hydrochlorofluorocarbons (HCFCs) Included in the Kigali Amendment to the Montreal Protocol"

_Atmospheric Chemistry and Physics, 2018_

## Referee Comment (RC1) · Anonymous Referee #1 · 7 Feb 2018

Review of Manuscript "Global Warming Potentials for the $C_1$-$C_3$ Hydrochlorofluorocarbons (HCFCs) Included in the Kigali Amendment to the Montreal Protocol," by D.K. Papanastasiou et al.

The paper is a study by Burkholder et al. to synthetically estimate atmospheric rate constants (using certain models and approximations to estimate the atmospheric loss processes), as well as to use (*ab initio*) synthetic calculated infrared absorption spectra so as to ultimately derive estimates for the lifetimes and global warming potentials for a series of the C1-C3 hydrochlorofluorocarbons contained in the Kigali amendment to the Montreal protocol.

In that amendment, when accounting for all of the C1 to C3 conformers of the HCFCs, there are 274 species of interest, but of these only 15 have both the actual measured relevant kinetic rate constants and requisite quantitative infrared absorption spectra to derive accurate values for the RE and ultimately the associated GWP. It will take many years and more likely many decades until most or all of the experimental data are acquired for these 274 species, so any efforts to derive some interim values using estimates, models and theoretical calculations are clearly justified. Such is the motivation for the present work. The paper thus represents a seminal contribution for estimating the GWP values for these 259 species; it also represents a very great deal of work. To write such a paper requires significant expertise in kinetics, atmospheric chemistry and modelling, IR spectroscopy, quantum chemical calculations, etc. The paper should clearly be published. We thus recommend publication with only minor modification.

This reviewer does have concerns leading us to criticise certain aspects of the study. Without pointing to a specific paragraph, the overall hue or colour of the narrative gives the reader an impression of the resultant data or GWP values indicate greater precision and accuracy than is perhaps warranted: The phrase "… such estimates are accurate enough for our study" is used quite often. The paper is in essence a (warranted) theoretical study that of necessity involves many approximations or extrapolations to derive certain data which are used with other data to calculate the lifetime and GWP values. Many of these data have non-trivial uncertainties and represent possible sources of significant error. However, in several places phrases to the effect "this study provides reliable policy relevant GWP metrics" imply significant accuracy. For example, inspection of column 5 in their Table 2 manifests their own estimates as the 100-year GWP values with ranges that at first blush appear to vary from 5x to 35x between the lower and upper bounds! While the data are clearly needed, the language should perhaps be modified to recognize the limitations of such calculated data.

Related to this phraseology aspect is the need for their intro/summary text to point out that more and better experimental data of several different kinds are needed: 1) Experimental measurements of atmospheric lifetimes are clearly needed; the laboratory of Dr. Burkholder and others are known for careful measurement of such rate constants and these are clearly warranted – while the SARs are good approximations, actual experimental data are required. 2) Even for relatively simple molecules such as the smaller HFCFs, the *ab initio* Gaussian calculations still can vary significantly from the measured infrared cross sections (see below), and these IR data need to be measured experimentally as well, including to longer wavelengths as the authors point out. Current experimental IR absorption data can have accuracies of 3% or better. 3) Whilst measurement of such kinetic and IR absorption data for all of the 259 named molecules will likely not occur soon, even increasing the number of

species in the training set (Table 1) from 15 to 25 or 35 or ? would obviously increase the accuracy and relevancy of the study. Such considerations need to be suggested in either the introduction or discussion sections, or both. A more appropriate slant to be taken throughout the manuscript would thus be words to the effect: "In the absence of such experimental data, the following study provides estimates of…"

One other general comment is that in several spots the paper would do better to simply flesh out some of the details as to how the results and approximations are realised. This is especially true for the experimental and calculation sections. However, their error analysis is likely the best approach for a study of this magnitude.

Finally, the section of the manuscript that deserves a bit more attention is section 2.3 where the intensities of the calculated IR spectra are compared to the measured IR spectra over the spectral region 2000-500 cm$^{-1}$. One initial question is how do the computed LWIR intensities compare with known values? Moreover, while not an expert in the Gaussian calculations, this reviewer fears that the 20% estimate as to the uncertainty in the IR absorbance spectra may perhaps be a bit too generous. The vibrational frequencies can now routinely be calculated with an accuracy of ca. 1 to 2% using such a basis set, and this is sufficient except in those cases where the predicted line frequency (depending on accuracy) is either adjacent to or obscured by an atmospheric absorption bands due to $H_2O$, $CO_2$ or $O_3$. However, the IR absorption intensities are far more difficult to predict with such accuracy - the eigenvalues (frequencies) to the nuclear motion equation are far less susceptible to approximations in the wavefunction than are the vibrational intensities (the eigenvectors). Besides the 20% estimate obtained from the training set, a more extensive discussion of the accuracy of the Gaussian results is likely justified.

The authors have used DFT methods (B3LYP/6-31G(2df,p)) to estimate total IR cross sections, and have compared their results with a training set of molecules, to show that they are confident that their estimated cross sections are within about 20% of measured cross sections, for those that have experimental values. Their claimed accuracy would be made stronger if they had also compared their B3LYP method with a few standard small molecules that have been used to assess absolute accuracy of computed IR cross sections, as in Jemenez-Hoyos et al., PCCP, 2008, 6621-6629, and Halls and Schlegel, J Chem Phys, 1998, vol 109, 10587-10593, where the standard deviation of B3LYP computed IR cross section (by individual band) is shown to be about 11 km/mol (conversion to cross section is certainly known by the authors) for a set of molecules whose band strengths extend from less than 10 km/mol to about 90 km/mol. In addition to these, more recent work has shown that for a set of alkanes, the absolute errors for IR band strengths were shown to be less than about 15% for B3LYP/6-31+G(d,p) computation compared to experiment, Williams et al., JQSRT, 2013, vol 129, 298-307.

It would be beneficial for the authors to review / cite some papers that have discussed the accuracy of the IR vibrational intensities, especially those that are conformer dependent. [A quick review of the literature pulled up the references listed below.] A similar discussion as to "quick and dirty estimate" may also be justified for two other areas: i) the neglect of the [37]Cl isotopomers and ii) the SAR relationships used, as discussed on p. 7., lines 110 to 114.

**Minor Suggestions / Errata**

Title: Perhaps insert "Estimates" into the title to read: "Global Warming Potential Estimates for the C1-C3 Hydrochlorofluorocarbons (HCFCs)…"

Pg. 3 line 61: Perhaps change first sentence to read "Due to the current dearth of such experimental data, the objective of the present work is to provide a best-effort and comprehensive evaluation…"

p. 4. Line 63: Define the acronym GTP. 1, sent.21: In previous studies, emission factors have units of $g\,kg^{-1}$, here the emission factors have units of $g\,km^{-1}$. Please explicitly define the emission factor that you are estimating somewhere in the manuscript.

p. 7 line 120: Insert "red" between "solid" and "symbols"

p. 8 line 134 – 148. Almost every sentence in these two paragraphs mentions another source of error. Can the composite uncertainty be estimated at the end of this section 2.1?

p. 8 line 153: It would be well to not only define FRF, but explain its meaning

p. 10 line 206. Is the comparison of experimental and predicted IR spectrum a "typical" result or is this one of the better matches?

p. 11 Figure 3: Where do the experimental data come from in this Figure? One of the references?

p. 11 Figure 3: The figure could facilitate a better comparison of predicted v. measured intensities if the y-axis were equally stretched in the two frames such that they were at the same factor, e.g. that 2.0 E-18 units were the same length in the top and bottom panels. While quite weak, it appears there are two small bands in the experimental data near ca. 610 and ca. 1385 $cm^{-1}$ that are missed in the Gaussian calculations.

p. 12 Figure 4: It would be helpful to put the units ($cm^2$ molecule$^{-1}$ $cm^{-1}$) directly on the plot of this figure, not just in the legend.

p. 14 line 294: This sentence seems perhaps unjustified: While there are far fewer data for bands below 500 $cm^{-1}$, it is curious that "the contribution of vibrational bands in this region to the RE is… usually minor, i.e. <1%". While it is true there are often fewer fundamental below 500 $cm^{-1}$, the ability to act as a greenhouse gas is also a function of the blackbody radiation of the earth and the blackbody curve near 295K maximizes near 1000 $cm^{-1}$, with very appreciable intensity from 200 to 1000 $cm^{-1}$. This sentence needs clarification or a reference to more extensive work. It may also suggest the need for more experimental measurements of IR intensities in the far-infrared.

p. 17 Figure 6: Good Figure-lots of information, which is well presented.

p. 18 line 356: Establishing the reliability of the metrics based on the average behaviour can be quite an approximation. Perhaps include language that acknowledges this approximation.

p. 18 line 370: This statement seems a bit skeptical and perhaps further explanations are needed to convince the readers that indeed the computed REs are within 10% of experimental values. Furthermore, a discussion of the estimated band strength uncertainty may strengthen the validity of the statement.

p. 18 line 375-377: Similar to the comment above, this sentence needs to be justified and it may be beneficial to have a discussion regarding the accuracy of the Gaussian results somewhere in the manuscript.

p. 18 line 378: This is not true for the spectral region 700-1300 $cm^{-1}$.

p. 19 line 382: It might be valuable to emphasize or to provide additional details regarding this phenomenon.

p. 19 line 402-403: This sentence may benefit from some clarification. When compared to results from Betowski et al., it appears that the present results are 29% greater, which would suggested that the accuracy of the present results is at least ≥29%. There are many uncertainties presented in this section. It might be helpful to the readers to have a table displaying all of the uncertainty values in the SI.

p. 19 line 409-411: Might be beneficial to include how the present results compare to other experiments. Including more studies may strengthen this section of the manuscript.

p. 20 line 425: The term "policy-relevant" is used in this sentence, yet the present results are off by a factor of 12. Again, adding language that acknowledges the limitations of such calculated data and the need for additional experimental data is likely justified.

p. 22 line 457: The need for additional laboratory studies is absolutely needed, and we suggest this point be emphasized throughout the paper.

Papers Discussing Accuracies of Calculated Gaussian Intensities, esp. for Multiple Conformers

Halls, M. D.; Schlegel, H. B., Comparison of the performance of local, gradient-corrected, and hybrid density functional models in predicting infrared intensities. *The Journal of chemical physics* **1998,** *109* (24), 10587-10593.

Jiménez-Hoyos, C. A.; Janesko, B. G.; Scuseria, G. E., Evaluation of range-separated hybrid density functionals for the prediction of vibrational frequencies, infrared intensities, and Raman activities. *Physical Chemistry Chemical Physics* **2008,** *10* (44), 6621-6629.

Lindenmaier, R.; Williams, S. D.; Sams, R. L.; Johnson, T. J., Quantitative Infrared Absorption Spectra and Vibrational Assignments of Crotonaldehyde and Methyl Vinyl Ketone Using Gas-Phase Mid-Infrared, Far-Infrared, and Liquid Raman Spectra: s-cis vs s-trans Composition Confirmed via Temperature Studies and ab Initio Methods. *The Journal of Physical Chemistry A* **2017,** *121* (6), 1195-1212.

Williams, S. D.; Johnson, T. J.; Sharpe, S. W.; Yavelak, V.; Oates, R.; Brauer, C. S., Quantitative vapor-phase IR intensities and DFT computations to predict absolute IR spectra based on molecular structure: I. Alkanes. *Journal of Quantitative Spectroscopy and Radiative Transfer* **2013,** *129*, 298-307.

---

## Referee Comment (RC2) · Anonymous Referee #2 · 1 Mar 2018

This paper describes the derivation of atmospheric lifetimes, ozone depletion potentials, infrared spectra, radiative efficiencies and global warming and temperature potentials for a very comprehensive set of HCFCs. The work is largely based on theoretical approaches but has a strong focus on comparisons with observation- and lab-based data. The paper is generally of sufficient quality and novelty for publication in ACP. I do however have two main concerns: Firstly, previous literature on theoretical calculations of GWPs, ODPs, etc. is largely ignored. Including more references at least for the most important HCFCs would also help the authors to highlight why their approach is

superior to previously published works. Secondly, the authors calculate ODPs partly based on outdated values as is described in one of the specific comments below.

Title: I think the current title describes the content of the paper insufficiently.

Figure S1: HFC-227ea is misspelled and HCFC-22 and HFC-125 appear twice.

Page 5, line 15-16: This is misleading as only one HCFC seems to have been used.

Page 5, line 16-17 and line 19: What does 'in most cases' mean? 51 %?

Page 6, line 14-15: The method previously used for calculating fractional release has recently been proved wrong and age-of-air estimates have been improved, both of which have substantial implications for a number of compounds including HCFCs. I am surprised that the editor did not question this as he is an author on all three recent papers (Ostermoeller et al., 2017; Engel et al., 2017; Elvidge et al., accepted, 2018 – all ACP).

Figure 3: It would help to see the experimental and the calculated spectrum in the same plot.

Page 11, line 29-31: I don't think there should be a section for GTPs if it only contains one sentence.

Page 17, line 3-4: Why are these HCFCs of primary interest?
* * *

---

## Referee Comment (RC3) · Anonymous Referee #3 · 2 Mar 2018

General comments:

The paper by Papanastasiou et al. provides estimates of lifetimes, ODPs, REs, GWPs and GTPs for a large number of HCFCs. The study is comprehensive and provides estimates that are very relevant for the recent Kigali Amendment to the Montreal Protocol. Although I recommend publication of the paper, there are some issues that need to be addressed first. Please see detailed comments below.

Specific comments:

[Figure]

Abstract: It would be good to include some of the results in the abstract. E.g., give the range of lifetimes and GWP 100-year values.

Page 1, line 23: "Reliable" is too strong in my opinion, considering that the difference from experimentally-derived values can be quite large for some compounds (as shown in Fig. S3).

Page 2, line 1-2: The sentence looked a bit strange to me. Perhaps better with "an exemption for countries with high ambient temperature"?

Page 2, line 17: Please change "global temperature potentials" to "global temperature change potentials" throughout the manuscript.

Introduction: There are hardly any references to previous work, although I know a lot of work has been done on the topic of calculating absorption spectra and resulting metrics. I do not ask for a review of previous work, but some introduction to the topic on calculated vs. experimental spectra should be included. I also suggest to add references to GWP, ODP and GTP on first use, as all readers may not be familiar with all the terms.

Table 1: Where is the IR absorption spectrum for HCFC-123a from? For many of the compounds, absorption spectra are available from several sources (see Table 4 in Hodnebrog et al., 2013). What is the reason for using absorption spectra from (in most cases) only one of the sources? Would be good to briefly state that. Also, in footnote 2 the terms lifetime-adjustment and stratospheric temperature correction have not been defined and could therefore seem confusing for readers not familiar with these. I suggest referring to the appropriate method section where these terms are explained.

Page 5, line 4: As I understand it, these are comparisons to experimental data. I suggest changing to "... for the training dataset with experimental rate coefficients...", just to make that clear.

Page 7, line 14-16: Is this shown somewhere? If not, adding "(not shown)" to the end of the sentence would be clarifying.

Page 8, line 5-6: Differences look larger than 2% in Fig. 3, especially for the band around 1100-1200 cm-1.

Figure 3: It would be much easier to compare the calculated vs. experimental spectra if they were in the same plot.

Page 10, line 9-11: Perhaps I missed something, but is it shown somewhere that the broadening leads to better agreement with experimental HCFC spectra? As I interpret Fig. 5, it only shows the difference with and without the broadening and not comparison to experimental data.

Page 10, line 23: I cannot see that Figure 5 includes all HCFCs studied, when compared to Table 2.

Figure 5: I think there is something wrong with the labeling above the plots - compounds HCFC-224 to HCFC-233 are listed twice. A minor point is that it would be more natural to switch the order of the plots, since the broadening sensitivity is discussed first.

Page 11, line 15: Could you include "(see Section 5)" at the end of the sentence? I started looking for the datasheets in the supplementary information without finding it, before I realized these were only available on a web site.

Page 11, line 20: Isn't the IntRF_CO2(T) the integrated radiative forcing of CO2? Also, M_HCFC in the formula is not defined, I think.

Page 11, line 24-25: In my opinion, Figure S3 is important enough to be in the main manuscript instead of the supplementary. In addition it would be good with a table or figure comparing the calculated REs with those from the training dataset.

Page 11, line 29-31: The section on GTP is very short. I suggest to merge it with

section 2.5?

Page 14, line 18-19: Where is the comparison of REs between calculated and experimental section shown?

Table 2: I suggest stating that the range in GWP100 values is due to different isomers, so that the range is not misinterpreted as uncertainty due to the method.

Technical corrections:

Page 11, line 21: "as described above" -> "as described in Section 2.1" ?

Page 12, line 2: "stratosphere-adjusted" -> "stratospheric temperature adjusted"

Page 15, line 15: "didn't" -> "did not"

Supplementary Fig. S1: "Burkholder et al." is listed twice in the caption.

---

## Author Comment (AC1) · 5 Apr 2018

**Reviewer #1**

We thank the reviewer for their helpful and constructive comments. Our responses and revisions to the manuscript are outlined in detail below.

**Author General Comment:** Several of the reviewer comments refer to the estimated uncertainties presented in our manuscript. Estimating uncertainties in the atmospheric metrics, although important, has generally been ignored in the literature to date. However, we believe that uncertainties should be a part of the discussion. A major hurdle in such a discussion, however, is that it is difficult to quantitatively estimate uncertainties in estimated quantities. In our manuscript, we have taken the approach of estimating uncertainties based on the performance obtained using the HCFC training dataset, i.e., what we call average behavior. We acknowledge that not all molecules follow "average" behavior, but the intention is that our analysis will provide a general perspective on the issue of metric uncertainty. Note that we emphasize that direct fundamental laboratory studies are preferred. However, in the absence of laboratory data theoretical methods can provide reasonably reliable results.

There was a misinterpretation by this reviewer of the range of GWPs reported in Table 2 as the uncertainty range of our analysis. The reported range is indeed large, which reflects the fact that the isomers of a HCFC family have different properties, while the uncertainties for the individual isomers is estimated to be much less, ~30-50%. This misinterpretation represents a thread for several of the other reviewer comments. We have revised the text in an attempt to avoid this misinterpretation by others.

A general comment from this and other reviewers was a lack of a historical perspective and citations for the development of applying theoretical methods to the quantitative calculation of infrared spectra, which was a major component of our study. To address this issue, we have added text to Section 2.3 that is relevant to the development of the theoretical methods and have cited some of the previous applications from our laboratory as well as from others as examples.

**Reviewer Comment**: The paper is a study by Burkholder et al. to synthetically estimate atmospheric rate constants (using certain models and approximations to estimate the atmospheric loss processes), as well as to use (*ab initio*) synthetic calculated infrared absorption spectra so as to ultimately derive estimates for the lifetimes and global warming potentials for a series of the C1-C3 hydrochlorofluorocarbons contained in the Kigali amendment to the Montreal protocol.

> **Author Comment**: A minor point, the theoretical methods in our work used density functional theory (DFT) not *ab initio* methods.

**Reviewer Comment**: In that amendment, when accounting for all of the C1 to C3 conformers of the HCFCs, there are 274 species of interest, but of these only 15 have both the actual measured relevant kinetic rate constants and requisite quantitative infrared absorption spectra to derive accurate values for the RE and ultimately the associated GWP. It will take many years and more likely many decades until most or all of the experimental data are acquired for these 274 species, so any efforts to derive some interim values using estimates, models and theoretical calculations are clearly justified. Such is the motivation for the present work. The paper thus represents a seminal contribution for estimating the GWP values for these 259 species; it also represents a very great deal of work. To write such a paper requires significant expertise in

kinetics, atmospheric chemistry and modelling, IR spectroscopy, quantum chemical calculations, etc. The paper should clearly be published. We thus recommend publication with only minor modification.

> **Author Comment**: We truly appreciate the reviewer comments. We would like to clarify, however, that the 274 species is the total number of "isomers" for the C1-C3 HCFCs. Each isomer may have a number of "conformers" (i.e., molecular configurations with slightly higher energy), that were included in our metric determinations. We considered conformers within 2 kcal/mole of the lowest energy geometry, which enables us to account for at least ~98% of the Boltzmann population distribution.

**Reviewer Comment**: This reviewer does have concerns leading us to criticise certain aspects of the study. Without pointing to a specific paragraph, the overall hue or colour of the narrative gives the reader an impression of the resultant data or GWP values indicate greater precision and accuracy than is perhaps warranted: The phrase "… such estimates are accurate enough for our study" is used quite often. The paper is in essence a (warranted) theoretical study that of necessity involves many approximations or extrapolations to derive certain data which are used with other data to calculate the lifetime and GWP values. Many of these data have non-trivial uncertainties and represent possible sources of significant error. However, in several places phrases to the effect "this study provides reliable policy relevant GWP metrics" imply significant accuracy. For example, inspection of column 5 in their Table 2 manifests their own estimates as the 100-year GWP values with ranges that at first blush appear to vary from 5x to 35x between the lower and upper bounds! While the data are clearly needed, the language should perhaps be modified to recognize the limitations of such calculated data.

> **Author Response**: The reviewer has misinterpreted the range of values for the isomer GWPs presented in column 5 of Table 2 as the uncertainties associated with a given HCFC chemical formula. Second, the reviewer makes a reasonable point that some of the uncertainty statements in our paper are not quantitative. We have made an effort in this manuscript to convey to the reader that the metrics reported in this work are estimates that have uncertainty associated with them (the consideration of uncertainties in this type work is an issue often overlooked, or ignored, in previous studies and assessments). That said, it is not straightforward to assign accurate statistical uncertainties for molecules with "unknown" properties. However, we should be more careful in using phrases such as "suitable for this study", which was meant to imply that these values represent a minor uncertainty in comparison with other sources of uncertainty. In our approach, we have recommended "average" uncertainties based on our comparisons with literature data, i.e., our training dataset. Note that metrics derived from accurate laboratory measurements still will lead to metric uncertainty, which can be substantial, due to the methods used to determine atmospheric lifetimes and radiative properties. For example, the well-mixed radiative efficiency estimation method given in Hodnebrog et al. (2013) is estimated to have a 25% uncertainty associated with the method. These other sources of uncertainty were not included in our analysis.
>
> **Author Action**: We have removed the subjective "reliable" in several places as follows:
> *In the Abstract*: The results from this study provide  policy relevant GWP metrics for the HCFCs included in the Montreal Protocol in the absence of experimentally derived metrics.

*In the Introduction*: The objective of the present work is to provide a  comprehensive evaluation of the atmospheric lifetimes, ozone depletion potentials (ODPs), GWPs, and global temperature change potentials (GTPs) for the HCFCs listed in Annex C of the amended Protocol.

*Section 2.3*: Similar approaches have been used in earlier studies for other classes of molecules with  good results, see Hodnebrog et al. (2013) and references cited within.

*Section 3.1*: This method of RE determination is, therefore, expected to provide  good estimates of REs in the absence of experimentally based determinations.

*Summary*: Although this work has provided  a comprehensive set of estimated  metrics for the $C_1$-$C_3$ HCFCs that presently do not have experimental data, careful direct fundamental laboratory studies of an intended HCFC would better define the critical atmospheric loss processes (reaction and UV photolysis) used to evaluate atmospheric lifetimes.

We have revised the use of "accurate" as follows:

*Section 2.1*: Therefore, this method of accounting for stratospheric loss  leads to only a minor uncertainty in the calculated global lifetime.

*Section 2.1*: As shown later, the O($^1$D) reaction is a minor loss process, <1%, for nearly all the HCFCs included in this study and, therefore, the estimation method used is not critical as this loss process is a minor contributor to the global lifetime

**Reviewer Comment**: Related to this phraseology aspect is the need for their intro/summary text to point out that more and better experimental data of several different kinds are needed: 1) Experimental measurements of atmospheric lifetimes are clearly needed; the laboratory of Dr. Burkholder and others are known for careful measurement of such rate constants and these are clearly warranted – while the SARs are good approximations, actual experimental data are required. 2) Even for relatively simple molecules such as the smaller HFCFs, the *ab initio* Gaussian calculations still can vary significantly from the measured infrared cross sections (see below), and these IR data need to be measured experimentally as well, including to longer wavelengths as the authors point out. Current experimental IR absorption data can have accuracies of 3% or better. 3) Whilst measurement of such kinetic and IR absorption data for all of the 259 named molecules will likely not occur soon, even increasing the number of species in the training set (Table 1) from 15 to 25 or 35 or ? would obviously increase the accuracy and relevancy of the study. Such considerations need to be suggested in either the introduction or discussion sections, or both. A more appropriate slant to be taken throughout the manuscript would thus be words to the effect: "In the absence of such experimental data, the following study provides estimates of…"

> **Author Response**: We agree with the reviewer sentiment whole heartedly. For compounds of interest direct careful laboratory studies are surely preferred. We have expressed this point of view in both the introduction and conclusion sections of our paper. It is also implicit in our approach that an improved (expanded) training dataset would be beneficial to the analysis presented in our work.
>
> **Author Action**: None

**Reviewer Comment**: One other general comment is that in several spots the paper would do better to simply flesh out some of the details as to how the results and approximations are realised. This is especially true for the experimental and calculation sections. However, their error analysis is likely the best approach for a study of this magnitude.

> **Author Response**: The results and approximations used in our work are based on the behavior of the training dataset, which for HCFCs is rather limited. We present our estimated uncertainties based on the "average" behavior (as discussed above) of the training dataset, although outliers are acknowledged. Note that the Betowski et al. study used a much larger training dataset that included a variety of classes of molecules to derive general properties that were applied to HCFCs. We argue that this approach resulted in a bias in their HCFC results (note that their theoretical results and those presented in this work are in good agreement).
>
> **Author Action**: None

**Reviewer Comment**: Finally, the section of the manuscript that deserves a bit more attention is section 2.3 where the intensities of the calculated IR spectra are compared to the measured IR spectra over the spectral region 2000-500 cm$^{-1}$. One initial question is how do the computed LWIR intensities compare with known values? Moreover, while not an expert in the Gaussian calculations, this reviewer fears that the 20% estimate as to the uncertainty in the IR absorbance spectra may perhaps be a bit too generous. The vibrational frequencies can now routinely be calculated with an accuracy of ca. 1 to 2% using such a basis set, and this is sufficient except in those cases where the predicted line frequency (depending on accuracy) is either adjacent to or obscured by an atmospheric absorption bands due to $H_2O$, $CO_2$ or $O_3$. However, the IR absorption intensities are far more difficult to predict with such accuracy - the eigenvalues (frequencies) to the nuclear motion equation are far less susceptible to approximations in the wavefunction than are the vibrational intensities (the eigenvectors). Besides the 20% estimate obtained from the training set, a more extensive discussion of the accuracy of the Gaussian results is likely justified.

> **Author Response**: A thorough discussion of the uncertainties associated with theoretically calculated infrared absorption spectra is relevant to this work, and that of others, but is well beyond the scope of our manuscript. Because of the volume of calculations required in this work, we have used DFT methods that yield reasonable results without too much computing cost. We have based our uncertainty estimates on how well the DFT methods work for the training dataset. Future studies could build on the work presented in this manuscript. We have added text to the beginning of Section 2.3 to provide better perspective for the calculation methods (text given below). The next comment is related to this one.

**Reviewer Comment**: The authors have used DFT methods (B3LYP/6-31G(2df,p)) to estimate total IR cross sections, and have compared their results with a training set of molecules, to show that they are confident that their estimated cross sections are within about 20% of measured cross sections, for those that have experimental values. Their claimed accuracy would be made stronger if they had also compared their B3LYP method with a few standard small molecules that have been used to assess absolute accuracy of computed IR cross sections, as in Jemenez-Hoyos et al., PCCP, 2008, 6621-6629, and Halls and Schlegel, J Chem Phys, 1998, vol 109, 10587-10593, where the standard deviation of B3LYP computed IR cross section (by individual band) is shown to be about 11 km/mol (conversion to cross section is certainly known by the authors) for a set of molecules whose band strengths extend from less than 10 km/mol to about 90 km/mol. In addition

to these, more recent work has shown that for a set of alkanes, the absolute errors for IR band strengths were shown to be less than about 15% for B3LYP/6-31+G(d,p) computation compared to experiment, Williams et al., JQSRT, 2013, vol 129, 298-307.

It would be beneficial for the authors to review / cite some papers that have discussed the accuracy of the IR vibrational intensities, especially those that are conformer dependent. [A quick review of the literature pulled up the references listed below.] A similar discussion as to "quick and dirty estimate" may also be justified for two other areas: i) the neglect of the 37Cl isotopomers and ii) the SAR relationships used, as discussed on p. 7., lines 110 to 114.

> **Author Response**:  We had not reviewed the history of theoretical calculations and the application to molecules of atmospheric interest in our original submission.  The other reviewers have also requested that this be included in our revised manuscript along with citation of some application papers.
>
> **Author Action**:  We have added "new" text with ample literature citations at the beginning of Section 2.3 "Theoretical Calculations".  The text includes the papers suggested by the reviewer as well as additional relevant material and some examples of work from our lab and others.  We have also added the following text to address not including calculations with Cl-atom isotopes (note that including all permutations of Cl-atom isotopes would have expanded the depth of the present work tremendously, while not altering the general conclusions from this work): "In principle, substitution of $^{35}$Cl by $^{37}$Cl in a heavy molecule would lower the frequency of the C-Cl stretch by ~3%".

**Text added to the start of section 2.3:**

Information about molecular vibrational frequencies, central to the interpretation of infrared spectra, thermodynamics, and many other aspects of chemistry, became amenable to computational determination in the early 1980s.  It was recognized that computed harmonic frequencies derived via the second derivative of energy as a function of atomic position were systematically higher than observed fundamentals and scale factors were introduced (Hout et al., 1982; Pople et al., 1981).  For Hartree-Fock frequencies these were typically ~0.9 and accounted both for the influence of anharmonicity and deficiencies in the underlying quantum calculations.  Frequencies based on methods incorporating electron correlation such as CCSD, CCSD(T) or certain functionals within density functional theory (DFT) often perform well for harmonic frequencies and are scaled by ~0.95 to match fundamental vibrational modes.  Such scaling has been updated as more methods appear (Alecu et al., 2010; Scott and Radom, 1996).  Rather less information is available concerning the evaluation of absorption intensities for fundamental modes.  Within the same harmonic approximation, implemented in popular quantum codes, the intensity is proportional to the square of the derivative of the dipole moment with respect to position.  Halls and Schlegel evaluated QCISD results against experiment and their plot indicates deviations of up to around ±20% and then used QCISD as a benchmark to evaluate a range of functionals (Halls and Schlegel, 1998).  For B3LYP, they found differences from QCISD of around 10%.  More recently, tests of the B3LYP functional found good performance for frequency and intensity (Jiménez-Hoyos et al., 2008; Katsyuba et al., 2013).  Some prior work where similar methods have been applied to the infrared absorption for molecules of atmospheric interest include studies of fluoromethanes (Blowers and Hollingshead, 2009), unsaturated hydrofluorocarbons (Papadimitriou and Burkholder, 2016; Papadimitriou et al., 2008b), perfluorocarbons (Bravo et al., 2010), chloromethanes (Wallington et al., 2016), $SO_2F_2$ (Papadimitriou et al., 2008a),

permethylsiloxanes (Bernard et al., 2017), and large survey studies such as by Kazakov et al. (Kazakov et al., 2012) and Betowski et al. (Betowski et al., 2015) to name a few.

Halls and Schlegel noted that real spectra may exhibit the influences of resonances, intensity sharing, and large-amplitude anharmonic modes. These can be partially accounted for in an analysis based on higher derivatives of the energy and the dipole moment, performed for instance within the framework of second-order vibrational perturbation theory (Barone, 2005). Advantages include treatment of resonances among vibrational levels and incorporation of overtones and combination bands. Examples of applications to molecules containing C-H and C-F bonds indicate excellent accord with experiment for band position and intensity, (Carnimeo et al., 2013) but for $CH_2ClF$ the intensity in the region involving C-Cl stretching nevertheless exhibits intensity errors of ~10% (Charmet et al., 2013).

**Additional text added within the section**:

We are not aware of prior studies of infrared spectra of HCFC conformers, but there have been prior theoretical studies of the conformers of other classes of molecule, such as for validation of observed infrared spectra used to deduce relative energies of carbonyl conformations (Lindenmaier et al., 2017) and comparison with measured infrared intensities for linear alkanes (Williams et al., 2013). The different errors and their trends for the intensities of C-H stretching and HCH bending modes indicate that a simple scaling approach, so successful for frequencies, will not work for intensities.

**and**

In fact the intensities of C-Cl stretches are a long-known problem for calculation (Halls and Schlegel, 1998).

**Minor Suggestions / Errata**
**Reviewer Comment**: Title: Perhaps insert "Estimates" into the title to read: "Global Warming Potential Estimates for the C1-C3 Hydrochlorofluorocarbons (HCFCs)…"
    **Author Response**: Agee
    **Author Action:** Title changed as follows: Global Warming Potential Estimates for the $C_1$-$C_3$ Hydrochlorofluorocarbons (HCFCs) Included in the Kigali Amendment to the Montreal Protocol

**Reviewer Comment**: Pg. 3 line 61: Perhaps change first sentence to read "Due to the current dearth of such experimental data, the objective of the present work is to provide a best-effort and comprehensive evaluation…"
    **Author Response**: We have used the phrase the "in the absence of experimental data…" in our manuscript, which we believe is a concise and accurate description.
    **Author Action**: None

**Reviewer Comment**: p. 4. Line 63: Define the acronym GTP. 1, sent.21: In previous studies, emission factors have units of g kg-1, here the emission factors have units of g km-1. Please explicitly define the emission factor that you are estimating somewhere in the manuscript.
    **Author Response**: GTP is defined on its first use on page 2. We are not addressing or discussing emission factors in our manuscript. So, we don't understand the origin of this comment.

**Author Action**:  None

**Reviewer Comment**:  p. 7 line 120: Insert "red" between "solid" and "symbols"
> **Author Response**:  This refers to the caption for Figure 1.  Red is not necessary as there are only solid and open symbols on the graph.
> **Author Action**:  None

**Reviewer Comment**:  p. 8 line 134 – 148. Almost every sentence in these two paragraphs mentions another source of error. Can the composite uncertainty be estimated at the end of this section 2.1?
> **Author Response**:  There are estimated uncertainties in the lifetimes and calculated infrared absorption spectra that propagate into the metrics as discussed in this text. Therefore, each parameter is considered separately and then combined to obtain an estimated uncertainty in the metric at the end of the discussion.  Seeing that the uncertainties are estimated values, we feel that reporting uncertainty values in a table would place too much emphasis on the accuracy of these estimated values.
> **Author Action**:  None

**Reviewer Comment**:  p. 8 line 153: It would be well to not only define FRF, but explain its meaning
> **Author Response**: We have not included background material on the derivation or meanings of lifetime, ODP, RE, GWP, GTP, and FRF in our manuscript.  Instead, we have provided the pertinent references that provide the detail necessary to fully understand these metrics.
> **Author Action**:  None

**Reviewer Comment**:  p. 10 line 206. Is the comparison of experimental and predicted IR spectrum a "typical" result or is this one of the better matches?
> **Author Response**:  This comment refers to Figure 3.  This result is typical.  A comparison of the individual experimentally reported spectra and our calculated spectra for all molecules with experimental data is provided graphically in the SI.
> **Author Action**:  We have added the follow text following the introduction of Figure 3: "A comparison of experimental and theoretical spectra for all molecules with experimental data is provided in the SI (see Section 5).".

**Reviewer Comment**:  p. 11 Figure 3: Where do the experimental data come from in this Figure? One of the references?
> **Author Response**: This comment applies to both Figure 3 and 4.  The source of the experimental spectra are given in Table 1.
> **Author Action**:  We have reiterated the source of the experimental spectra in the captions for Figures 3 and 4 as follows: "… (see Table 1 for the source of the experimental spectrum)".

**Reviewer Comment**:  p. 11 Figure 3: The figure could facilitate a better comparison of predicted v. measured intensities if the y-axis were equally stretched in the two frames such that they were at the same factor, e.g. that 2.0 E-18 units were the same length in the top and bottom panels.

While quite weak, it appears there are two small bands in the experimental data near ca. 610 and ca. 1385 cm$^{-1}$ that are missed in the Gaussian calculations.

**Author Response**: The Gaussian calculation presented here considers only the fundamental vibration frequencies. Therefore, it is possible that weak combination or overtone bands are not included in the calculated spectra. The overall quality of the literature infrared absorption reference spectra was not explored as part of our study. The other reviewers have request that the panels for this figure be combined.

**Author Action**: Figure revised as follows:

[Figure]

**Reviewer Comment**: p. 12 Figure 4: It would be helpful to put the units (cm2 molecule-1 cm-1) directly on the plot of this figure, not just in the legend.

**Author Response**: Okay

**Author Action**: Units removed from figure caption and figure revised as follows:

[Figure]

**Reviewer Comment**: p. 14 line 294: This sentence seems perhaps unjustified: While there are far fewer data for bands below 500 cm$^{-1}$, it is curious that "the contribution of vibrational bands in this region to the RE is… usually minor, i.e. <1%". While it is true there are often fewer fundamental below 500 cm$^{-1}$, the ability to act as a greenhouse gas is also a function of the blackbody radiation of the earth and the blackbody curve near 295K maximizes near 1000 cm-1, with very appreciable intensity from 200 to 1000 cm$^{-1}$. This sentence needs clarification or a reference to more extensive work. It may also suggest the need for more experimental measurements of IR intensities in the far-infrared.

> **Author Response**:  Our results would indicate that an experimental focus on far-infrared absorption bands is not presently warranted for HCFCs.  This is because the band intensities in this region are usually much weaker than the C-F stretching region and our calculations show that the contribution to the RE would be small, less than 1% in nearly all cases.  Stated another way: we are guiding experimental effort away from an unproductive effort, because it is not easy in the laboratory to go below 500 cm$^{-1}$ in the spectrum measurements.
> **Author Action**:  None

**Reviewer Comment**:  p. 17 Figure 6: Good Figure-lots of information, which is well presented.
> **Author Response**: Thanks

**Reviewer Comment**:  p. 18 line 356: Establishing the reliability of the metrics based on the average behaviour can be quite an approximation. Perhaps include language that acknowledges this approximation.

> **Author Response**:  In general, we agree with this comment, but as discussed in response to other comments we do not feel that a more quantitative analysis is possible at this time for molecules lacking experimental data.
> **Author Action**:  None

**Reviewer Comment**:  p. 18 line 370: This statement seems a bit skeptical and perhaps further explanations are needed to convince the readers that indeed the computed REs are within 10% of experimental values. Furthermore, a discussion of the estimated band strength uncertainty may strengthen the validity of the statement.

> **Author Response**:  The estimated RE uncertainty given in our manuscript is based on the average behavior found for the training dataset, which shows roughly a ~10% uncertainty. Sources of uncertainty in the RE calculation arise from the calculated frequencies and band strengths.  Combining these with the irradiance profile leads to some cancelation of error, although that is difficult to quantify other than in comparison of the calculated REs with the experimentally determined values.  In our error discussion, we have used a 20% RE uncertainty.
> **Author Action**:  We have added a figure to the SI that shows the agreement between the lifetime corrected experimental and calculated REs for the training dataset (see below). The following text was added to Section 2.4: A comparison of the experimentally derived REs and the calculated values for the training dataset is given in Figure S3.  Note that comparing 'lifetime corrected" REs increases the spread in the correlation between the experimental and calculated values because of differences between experimental and

calculated lifetimes and the correction factor. The figure shows the 20% correlation range, which is used in the metric uncertainty analysis. A general discussion of the uncertainties associated with band intensity calculation is now included in the introductory paragraphs to Section 2.3.

[Figure]

**Figure S3**: Comparison of experimental and calculated "lifetime corrected" radiative efficiencies (REs) for the training dataset HCFCs. The solid line is the 1:1 correlation. The gray shaded region represents ±20% around the 1:1 line. Note that the spread in REs without applying the "lifetime correction" is ~±10%.

**Reviewer Comment**: p. 18 line 375-377: Similar to the comment above, this sentence needs to be justified and it may be beneficial to have a discussion regarding the accuracy of the Gaussian results somewhere in the manuscript.

    **Author Response**: We are basing our estimated uncertainties on the level of agreement with the training dataset values, which is pretty good. A comprehensive discussion of the uncertainties associated with the Gaussian calculations, more specifically, the calculations using DFT methods, is beyond the scope of this work. We have addressed this point in the revised Section 2.3 and in an earlier response regarding estimated uncertainties.

    **Author Action**: None

**Reviewer Comment**: p. 18 line 378: This is not true for the spectral region 700-1300 cm$^{-1}$.

    **Author Response**: This comment regards the discussion that a change in the vibrational band centers primarily impacts the RE through the overlap with the irradiance profile. What is given in the manuscript is correct. This and the next comment are related.

    **Author Action**: None

**Reviewer Comment**: p. 19 line 382: It might be valuable to emphasize or to provide additional details regarding this phenomenon.

    **Author Response**: This comment refers to the possibility that the metric uncertainty might be greater if an absorbing molecule's infrared spectrum was near the $CO_2$ or $H_2O$

absorption features.  This issue has been mentioned in the text, but is difficult to quantify in general terms.  This is a situation where laboratory studies would be needed.
**Author Action**:  None

**Reviewer Comment**:  p. 19 line 402-403: This sentence may benefit from some clarification. When compared to results from Betowski et al., it appears that the present results are 29% greater, which would suggested that the accuracy of the present results is at least ≥29%. There are many uncertainties presented in this section. It might be helpful to the readers to have a table displaying all of the uncertainty values in the SI.
> **Author Response**:  The reviewer has misinterpreted the results that are already included in the text.  We actually agree very well with the raw HCFC results from the Betowski et al. study.  The 29% difference is due to the band strength scaling factor developed in the Betowski et al. study that was based on an analysis of many classes of compounds.  We argue that their band strength analysis yielded biased results for the HCFC class and that the 29% scaling factor should not be applied.  This difference is, therefore, not representative of the uncertainty in the analysis.
> **Author Action**:  None

**Reviewer Comment**:  p. 19 line 409-411: Might be beneficial to include how the present results compare to other experiments. Including more studies may strengthen this section of the manuscript.
> **Author Response**:  We do not agree with this comment.  Basically, what the reviewer is suggesting is what was done in the Betowski et al. work.  We already compare with all HCFC experiments, so other experiments would mean other classes of compound.  We argue that this approach that was used in the Betowski et al. work led to a bias in the HCFC results.  So, the approach used in the present work that limits the training dataset to HCFCs is most appropriate.
> **Author Action**:  None

**Reviewer Comment**:  p. 20 line 425: The term "policy-relevant" is used in this sentence, yet the present results are off by a factor of 12. Again, adding language that acknowledges the limitations of such calculated data and the need for additional experimental data is likely justified.
> **Author Response**:  The present results are not off by a factor of 12.  We believe that this comment is based on the misinterpretation of the range of GWPs given in Table 2 as described in a previous comments and responses.
> **Author Action**:  None

**Reviewer Comment**:  p. 22 line 457: The need for additional laboratory studies is absolutely needed, and we suggest this point be emphasized throughout the paper.
> **Author Response**:  We agree that experimental studies are preferred over theoretically calculated or empirically derived values.  We have emphasized this point in the most visible Introduction and Conclusion sections.
> **Author Action**:  None

---

## Author Comment (AC2) · 5 Apr 2018

**Reviewer #2**

We thank the reviewer for their helpful and constructive comments. Our responses and revisions to the manuscript are outlined in detail below.

**Reviewer Comment**: This paper describes the derivation of atmospheric lifetimes, ozone depletion potentials, infrared spectra, radiative efficiencies and global warming and temperature potentials for a very comprehensive set of HCFCs. The work is largely based on theoretical approaches but has a strong focus on comparisons with observation- and lab-based data. The paper is generally of sufficient quality and novelty for publication in ACP. I do however have two main concerns: Firstly, previous literature on theoretical calculations of GWPs, ODPs, etc. is largely ignored. Including more references at least for the most important HCFCs would also help the authors to highlight why their approach is superior to previously published works. Secondly, the authors calculate ODPs partly based on outdated values as is described in one of the specific comments below.

> **Author Comment**: The history of using theoretical methods to estimate infrared absorption spectra, which are needed for GWP determinations, dates back decades. Blowers and co-workers provided some of the first studies that applied theoretical methods to greenhouse gases of atmospheric relevance. Other noteworthy studies are the comprehensive works of NIST (Kazakov et al., 2012) and Betowski et al. (2015). We have included Betowski et al. in our discussion, but we could add other citations to provide additional perspective. The focus of our paper is on the application of these proven methods in a comprehensive manner to a class of compounds of interest to the Montreal Protocol, but for which laboratory data are not available. The novelty of this work is in that we have comprehensively addressed the role of conformers, which has been routinely neglected in previous studies, in deriving relevant metrics. We have also made an attempt to address the uncertainties associated with the estimated metrics, another topic frequently neglected. We don't believe our work is necessarily superior to similar previous studies, but it does provide a comprehensive systematic study of a large number of HCFCs of interest to the Montreal Protocol. The results obtained in our work are provided in great detail in the available SI, such that future studies by other research groups can benefit from this work. The semi-empirical ODPs provided in this work use the model lifetime results presented in the SPARC lifetime report, which are currently recommended for atmospheric modeling. We have also applied an empirical formulation, based on the SPARC lifetime results, for the fractional release factors for the many HCFCs that do not have reported values.

> **Author Action**: We have added text to the manuscript in section 2.3 Theoretical Calculations to address the lack of history and citations in the original submission.

**Text added to the start of section 2.3:**

Information about molecular vibrational frequencies, central to the interpretation of infrared spectra, thermodynamics, and many other aspects of chemistry, became amenable to computational determination in the early 1980s. It was recognized that computed harmonic frequencies derived via the second derivative of energy as a function of atomic position were systematically higher than observed fundamentals and scale factors were introduced (Hout et al., 1982; Pople et al., 1981). For Hartree-Fock frequencies these were

typically ~0.9 and accounted both for the influence of anharmonicity and deficiencies in the underlying quantum calculations. Frequencies based on methods incorporating electron correlation such as CCSD, CCSD(T) or certain functionals within density functional theory (DFT) often perform well for harmonic frequencies and are scaled by ~0.95 to match fundamental vibrational modes. Such scaling has been updated as more methods appear (Alecu et al., 2010; Scott and Radom, 1996). Rather less information is available concerning the evaluation of absorption intensities for fundamental modes. Within the same harmonic approximation, implemented in popular quantum codes, the intensity is proportional to the square of the derivative of the dipole moment with respect to position. Halls and Schlegel evaluated QCISD results against experiment and their plot indicates deviations of up to around ±20% and then used QCISD as a benchmark to evaluate a range of functionals (Halls and Schlegel, 1998). For B3LYP, they found differences from QCISD of around 10%. More recently, tests of the B3LYP functional found good performance for frequency and intensity (Jiménez-Hoyos et al., 2008; Katsyuba et al., 2013). Some prior work where similar methods have been applied to the infrared absorption for molecules of atmospheric interest include studies of fluoromethanes (Blowers and Hollingshead, 2009), unsaturated hydrofluorocarbons (Papadimitriou and Burkholder, 2016; Papadimitriou et al., 2008b), perfluorocarbons (Bravo et al., 2010), chloromethanes (Wallington et al., 2016), $SO_2F_2$ (Papadimitriou et al., 2008a), permethylsiloxanes (Bernard et al., 2017), and large survey studies such as by Kazakov et al. (Kazakov et al., 2012) and Betowski et al. (Betowski et al., 2015) to name a few.

Halls and Schlegel noted that real spectra may exhibit the influences of resonances, intensity sharing, and large-amplitude anharmonic modes. These can be partially accounted for in an analysis based on higher derivatives of the energy and the dipole moment, performed for instance within the framework of second-order vibrational perturbation theory (Barone, 2005). Advantages include treatment of resonances among vibrational levels and incorporation of overtones and combination bands. Examples of applications to molecules containing C-H and C-F bonds indicate excellent accord with experiment for band position and intensity, (Carnimeo et al., 2013) but for $CH_2ClF$ the intensity in the region involving C-Cl stretching nevertheless exhibits intensity errors of ~10% (Charmet et al., 2013).

**Additional text added within the section**:

We are not aware of prior studies of infrared spectra of HCFC conformers, but there have been prior theoretical studies of the conformers of other classes of molecule, such as for validation of observed infrared spectra used to deduce relative energies of carbonyl conformations (Lindenmaier et al., 2017) and comparison with measured infrared intensities for linear alkanes (Williams et al., 2013). The different errors and their trends for the intensities of C-H stretching and HCH bending modes indicate that a simple scaling approach, so successful for frequencies, will not work for intensities.

**and**

In fact the intensities of C-Cl stretches are a long-known problem for calculation (Halls and Schlegel, 1998).

**Reviewer Comment**:  Title: I think the current title describes the content of the paper insufficiently.

> **Author Response**:  Reviewer #1 suggested including "estimates" in the title to help clarify the content of the manuscript.
>
> **Author Action:**  Title changed as follows: Global Warming Potential Estimates for the $C_1$-$C_3$ Hydrochlorofluorocarbons (HCFCs) Included in the Kigali Amendment to the Montreal Protocol

**Reviewer Comment**:  Figure S1: HFC-227ea is misspelled and HCFC-22 and HFC-125 appear twice.

> **Author Response**:  Thanks, there were typos in the labels.
>
> **Author Action**:  The labeling has been corrected as follows:

[Figure]

**Reviewer Comment**:  Page 5, line 15-16: This is misleading as only one HCFC seems to have been used.

> **Author Response**:  This section addresses the determination of stratospheric lifetimes that, as stated, includes 2 HCFCs (142b was not included in the fit) and 8 HFCs.  We are not sure how this was misinterpreted.
>
> **Author Action**:  None

**Reviewer Comment**:  Page 5, line 16-17 and line 19: What does 'in most cases' mean? 51 %?

> **Author Response**:  This statement was intended to mean the vast majority.
>
> **Author Action**:  For clarification, the text has been revised as follows:  The stratospheric loss via the OH reaction accounts for ~$\leq$5% of the total OH loss process for >95% of the HCFCs.

**Reviewer Comment**:  Page 6, line 14-15: The method previously used for calculating fractional release has recently been proved wrong and age-of-air estimates have been improved, both of which have substantial implications for a number of compounds including HCFCs. I am surprised that the editor did not question this as he is an author on all three recent papers (Ostermoeller et al., 2017; Engel et al., 2017; Elvidge et al., accepted, 2018 – all ACP).

**Author Response**:  It is not clear how the results from the cited recent papers would be applied to molecules with unknown lifetimes and no observational data.  The empirical approach used in this work to relate the fraction of a molecule removed in the stratosphere to its fractional release is a reasonable estimation method, recommended by several atmospheric modeling groups, for use in the semi-empirical ozone depletion potential (ODP) calculation.   The preferred approach to determine ODPs would be to use atmospheric model calculations for each of the 274 HCFCs included in the work to calculate the ODPs.  Such calculations were, however, considered beyond the scope of this study.

**Author Action**:  None

**Reviewer Comment**:  Figure 3: It would help to see the experimental and the calculated spectrum in the same plot.

**Author Response**:  The original figure was split into two panels to minimize congestion.  However, this reviewer, and another, suggested combining the panels.

**Author Action**:  The graph was revised as follows as requested.

[Figure]

**Reviewer Comment**:  Page 11, line 29-31: I don't think there should be a section for GTPs if it only contains one sentence.

**Author Response**:  Okay

**Author Action**: Sub-section title has been removed and GTP added to the preceding sub-section title. "**2.5  Global Warming and Global Temperature change Potentials**"

**Reviewer Comment**:  Page 17, line 3-4: Why are these HCFCs of primary interest?

**Author Response**:  These are the HCFCs identified by the parties as some of the most likely candidates to watch for future use.  However, this is not an official policy statement.

**Author Action**:  None

---

## Author Comment (AC3) · 5 Apr 2018

**Reviewer #3**

We thank the reviewer for their helpful and constructive comments. Our responses and revisions to the manuscript are outlined in detail below.

**General reviewer comments**:

The paper by Papanastasiou et al. provides estimates of lifetimes, ODPs, REs, GWPs and GTPs for a large number of HCFCs. The study is comprehensive and provides estimates that are very relevant for the recent Kigali Amendment to the Montreal Protocol. Although I recommend publication of the paper, there are some issues that need to be addressed first. Please see detailed comments below.

**Reviewer Comment**: Abstract: It would be good to include some of the results in the abstract. E.g., give the range of lifetimes and GWP 100-year values.

> **Author Response**: Agree
> **Author Action**: We have revised the text in the abstract as follows: "The $C_1$–$C_3$ HCFCs display a wide range of lifetimes (0.3 to 62 years) and GWPs (5 to 5,330, 100-year time horizon) dependent on their molecular structure and H-atom content of the individual HCFC.".

**Reviewer Comment**: Page 1, line 23: "Reliable" is too strong in my opinion, considering that the difference from experimentally-derived values can be quite large for some compounds (as shown in Fig. S3).

> **Author Response**: Okay
> **Author Action**: We have removed the subjective "reliable" in several places as follows:
> *In the Abstract*: The results from this study provide  policy relevant GWP metrics for the HCFCs included in the Montreal Protocol in the absence of experimentally derived metrics.
> *In the Introduction*: The objective of the present work is to provide a  comprehensive evaluation of the atmospheric lifetimes, ozone depletion potentials (ODPs), GWPs, and global temperature change potentials (GTPs) for the HCFCs listed in Annex C of the amended Protocol.
> *Section 2.3*: Similar approaches have been used in earlier studies for other classes of molecules with  good results, see Hodnebrog et al. (2013) and references cited within.
> *Section 3.1*: This method of RE determination is, therefore, expected to provide  good estimates of REs in the absence of experimentally based determinations.
> *Summary*: Although this work has provided  a comprehensive set of estimated  metrics for the $C_1$-$C_3$ HCFCs that presently do not have experimental data, careful direct fundamental laboratory studies of an intended HCFC would better define the critical atmospheric loss processes (reaction and UV photolysis) used to evaluate atmospheric lifetimes.

**Reviewer Comment**: Page 2, line 1-2: The sentence looked a bit strange to me. Perhaps better with "an exemption for countries with high ambient temperature"?

> **Author Response**: Agree

**Author Action**: Text changed as follows: " which are different for developed and developing countries with an exemption for countries with high ambient temperature . ".

**Reviewer Comment**: Page 2, line 17: Please change "global temperature potentials" to "global temperature change potentials" throughout the manuscript.
    **Author Response**: Okay
    **Author Action**: Changed in two places in the text and two places in the SI.

**Reviewer Comment**: Introduction: There are hardly any references to previous work, although I know a lot of work has been done on the topic of calculating absorption spectra and resulting metrics. I do not ask for a review of previous work, but some introduction to the topic on calculated vs. experimental spectra should be included. I also suggest to add references to GWP, ODP and GTP on first use, as all readers may not be familiar with all the terms.
    **Author Response**: Agree
    **Author Action**: We have added text to the introduction to Section 2.3 "Theoretical Calculations" that provides background to the methods applied in this work and also cites literature work that has applied these methods from our laboratory and others.
    **Text added to the start of section 2.3:**

Information about molecular vibrational frequencies, central to the interpretation of infrared spectra, thermodynamics, and many other aspects of chemistry, became amenable to computational determination in the early 1980s. It was recognized that computed harmonic frequencies derived via the second derivative of energy as a function of atomic position were systematically higher than observed fundamentals and scale factors were introduced (Hout et al., 1982; Pople et al., 1981). For Hartree-Fock frequencies these were typically ~0.9 and accounted both for the influence of anharmonicity and deficiencies in the underlying quantum calculations. Frequencies based on methods incorporating electron correlation such as CCSD, CCSD(T) or certain functionals within density functional theory (DFT) often perform well for harmonic frequencies and are scaled by ~0.95 to match fundamental vibrational modes. Such scaling has been updated as more methods appear (Alecu et al., 2010; Scott and Radom, 1996). Rather less information is available concerning the evaluation of absorption intensities for fundamental modes. Within the same harmonic approximation, implemented in popular quantum codes, the intensity is proportional to the square of the derivative of the dipole moment with respect to position. Halls and Schlegel evaluated QCISD results against experiment and their plot indicates deviations of up to around ±20% and then used QCISD as a benchmark to evaluate a range of functionals (Halls and Schlegel, 1998). For B3LYP, they found differences from QCISD of around 10%. More recently, tests of the B3LYP functional found good performance for frequency and intensity (Jiménez-Hoyos et al., 2008; Katsyuba et al., 2013). Some prior work where similar methods have been applied to the infrared absorption for molecules of atmospheric interest include studies of fluoromethanes (Blowers and Hollingshead, 2009), unsaturated hydrofluorocarbons (Papadimitriou and Burkholder, 2016; Papadimitriou et al., 2008b), perfluorocarbons (Bravo et al., 2010), chloromethanes (Wallington et al., 2016), $SO_2F_2$ (Papadimitriou et al., 2008a), permethylsiloxanes (Bernard et al., 2017), and large survey studies such as by Kazakov et al. (Kazakov et al., 2012) and Betowski et al. (Betowski et al., 2015) to name a few.

Halls and Schlegel noted that real spectra may exhibit the influences of resonances, intensity sharing, and large-amplitude anharmonic modes. These can be partially accounted for in an analysis based on higher derivatives of the energy and the dipole moment, performed for instance within the framework of second-order vibrational perturbation theory (Barone, 2005). Advantages include treatment of resonances among vibrational levels and incorporation of overtones and combination bands. Examples of applications to molecules containing C-H and C-F bonds indicate excellent accord with experiment for band position and intensity, (Carnimeo et al., 2013) but for $CH_2ClF$ the intensity in the region involving C-Cl stretching nevertheless exhibits intensity errors of ~10% (Charmet et al., 2013).

**Additional text added within the section**:

We are not aware of prior studies of infrared spectra of HCFC conformers, but there have been prior theoretical studies of the conformers of other classes of molecule, such as for validation of observed infrared spectra used to deduce relative energies of carbonyl conformations (Lindenmaier et al., 2017) and comparison with measured infrared intensities for linear alkanes (Williams et al., 2013). The different errors and their trends for the intensities of C-H stretching and HCH bending modes indicate that a simple scaling approach, so successful for frequencies, will not work for intensities.

**and**

In fact the intensities of C-Cl stretches are a long-known problem for calculation (Halls and Schlegel, 1998).

Reference to metrics included as follows: The infrared spectra are then combined with our estimated global atmospheric lifetimes to estimate the lifetime and stratospheric temperature adjusted radiative efficiency (RE), GWP, and GTP metrics (see IPCC (2013) and WMO (2014) assessments).

**Reviewer Comment**: Table 1: Where is the IR absorption spectrum for HCFC-123a from? For many of the compounds, absorption spectra are available from several sources (see Table 4 in Hodnebrog et al., 2013). What is the reason for using absorption spectra from (in most cases) only one of the sources? Would be good to briefly state that. Also, in footnote 2 the terms lifetime-adjustment and stratospheric temperature correction have not been defined and could therefore seem confusing for readers not familiar with these. I suggest referring to the appropriate method section where these terms are explained.

> **Author Response**: Typically, the infrared spectra reported from different laboratories are in pretty good agreement and not a source of large uncertainty. The infrared spectra for HCFCs -31, -123a, -132b, -234fb, and -243cc are presently not available in the open literature. These molecules are included in Table 1 because kinetic data is available. There are also a few molecules where infrared data are available, but not kinetic data. In cases where multiple infrared spectrum measurements are available, we have used the spectra we think most reliable, although we have not performed a critical analysis.
>
> **Author Action**: No change for infrared spectra comment. Citation to IPCC and WMO for terms now given in Introduction (see response above).

**Reviewer Comment**: Page 5, line 4: As I understand it, these are comparisons to experimental data. I suggest changing to "... for the training dataset with experimental rate coefficients...", just to make that clear.

> **Author Response**: This is a comparison of SAR calculated rate coefficients for the molecules in the training dataset with the available experimental values. Not all molecules in the training dataset have experimental rate coefficient data available (see Table 1).
>
> **Author Action**: Text revised as follows: Comparison of structure activity relationship (SAR) OH rate coefficients for the training dataset (Table 1) with rate coefficients recommended in Burkholder et al. (2015).

**Reviewer Comment**: Page 7, line 14-16: Is this shown somewhere? If not, adding "(not shown)" to the end of the sentence would be clarifying.

> **Author Response**: We are quoting the results from our test calculations and the results are given in the text. There is not really anything else to show, or not show.
>
> **Author Action**: None

**Reviewer Comment**: Page 8, line 5-6: Differences look larger than 2% in Fig. 3, especially for the band around 1100-1200 cm-1.

> **Author Response**: There are some discrepancies among individual band strengths, but the total integrated band strengths are in good agreement.
>
> **Author Action**: Text clarified as follows: The calculated spectrum is in good agreement with the experimentally measured spectrum with band positions and total integrated band strengths agreeing to within ~2%.

**Reviewer Comment**: Figure 3: It would be much easier to compare the calculated vs. experimental spectra if they were in the same plot.

> **Author Response**: The original figure was split into two panels to minimize congestion. However, this reviewer and another suggested combining the panels.
>
> **Author Action**: The graph was revised as follows as requested.

[Figure]

**Reviewer Comment**: Page 10, line 9-11: Perhaps I missed something, but is it shown somewhere that the broadening leads to better agreement with experimental HCFC spectra? As I interpret Fig. 5, it only shows the difference with and without the broadening and not comparison to experimental data.

  **Author Response**: Of course, experimental data don't require spectral broadening. Implicit to our discussion of broadening is that a more realistic representation of the actual infrared absorption spectrum should provide a more realistic evaluation of the radiative metrics. Figure 5 was included to illustrate the **sensitivity** of including broadening in our calculations (something ignored in many studies of this type). Note that this does not necessarily mean that including broadening leads to a more accurate metric in our work, although it probably does. Figure 5 shows that in most cases the sensitivity is on the order of 5% for the HCFCs included in this work. We emphasize that this is a sensitivity analysis (as labeled in the figure) not an uncertainty analysis.

  **Author Action**: None

**Reviewer Comment**: Page 10, line 23: I cannot see that Figure 5 includes all HCFCs studied, when compared to Table 2. Figure 5: I think there is something wrong with the labeling above the plots – compounds HCFC-224 to HCFC-233 are listed twice. A minor point is that it would be more natural to switch the order of the plots, since the broadening sensitivity is discussed first.

  **Author Response**: This figure and its labelling were in error.

  **Author Action**: The figure has been revised as follows:

[Figure]

**Reviewer Comment**: Page 11, line 15: Could you include "(see Section 5)" at the end of the sentence? I started looking for the datasheets in the supplementary information without finding it, before I realized these were only available on a web site.

  **Author Response**: Okay. It was necessary to place these files on a web site because they exceeded the memory limits for the journal SI.

  **Author Action**: Text added: "Well-mixed and lifetime-adjusted RE values are included in the Supporting Information datasheets (see Section 5)."

**Reviewer Comment**: Page 11, line 20: Isn't the IntRF_CO2(T) the integrated radiative forcing of CO2? Also, M_HCFC in the formula is not defined, I think.

  **Author Response**: Agree

**Author Action**: Text revised as follows: where $IntRF_{CO2}(T)$ is the underline{integrated} radiative forcing of $CO_2$ underline{and $M_{HCFC}$ is the HCFC molecular weight}.

**Reviewer Comment**: Page 11, line 24-25: In my opinion, Figure S3 is important enough to be in the main manuscript instead of the supplementary. In addition it would be good with a table or figure comparing the calculated REs with those from the training dataset.

    **Author Response**: The GWP values are already given in Table 1 and Table S1. A RE correlation figure has been added to the SI.

    **Author Action**: We have moved the GWP correlation figure into the main body of the manuscript, new Figure 6.

**Reviewer Comment**: Page 11, line 29-31: The section on GTP is very short. I suggest to merge it with section 2.5?

    **Author Response**: Okay.

    **Author Action**: Sub-section title has been removed and GTP added to the preceding sub-section title. "**2.5 Global Warming and Global Temperature change Potentials**"

**Reviewer Comment**: Page 14, line 18-19: Where is the comparison of REs between calculated and experimental section shown?

    **Author Response**: We did not show a plot of calculated vs experimental REs in our original submission because of the similarity to the infrared spectrum correlation.

    **Author Action**: The figure below that compares the lifetime corrected REs has been added to the SI. The following text was added to Section 2.4: underline{A comparison of the experimentally derived REs and the calculated values for the training dataset is given in Figure S3.}

[Figure]

**Figure S3**: Comparison of experimental and calculated "lifetime corrected" radiative efficiencies (REs) for the training dataset HCFCs. The solid line is the 1:1 correlation. The gray shaded region represents ±20% around the 1:1 line. Note that the spread in REs without applying the "lifetime correction" is ~±10%.

**Reviewer Comment**: Table 2: I suggest stating that the range in GWP100 values is due to different isomers, so that the range is not misinterpreted as uncertainty due to the method.

    **Author Response**: Agree. Reviewer #1 misinterpreted the reported range in values as the uncertainty, which makes this comment even more relevant.

    **Author Action**: Table title revised as follows: "The Annex C HCFC table provided in the Kigali amendment to the Montreal Protocol, where the range of 100-year time horizon global warming potentials (GWPs) obtained in this work for various HCFC isomers all with the chemical formula given in the first column is given in *italics* *".

**Reviewer Comment**: Page 11, line 21: "as described above" -> "as described in Section 2.1" ?

    **Author Response**: Agree

    **Author Action**: Text added

**Reviewer Comment**: Page 12, line 2: "stratosphere-adjusted" -> "stratospheric temperature adjusted"

    **Author Response**: Agree

    **Author Action**: Text revised.

**Reviewer Comment**: Page 15, line 15: "didn't" -> "did not"

    **Author Response**: Okay

    **Author Action**: Changed text to "did not".

**Reviewer Comment**: Supplementary Fig. S1: "Burkholder et al." is listed twice in the caption.

    **Author Response**: Agree

    **Author Action**: Authors name has been suppressed in citation.

---

## Author Response (AR2)

**Technical Corrections**
**Manuscript ACP-2018-27**
Global Warming Potential Estimates for the $C_1$-$C_3$ Hydrochlorofluorocarbons (HCFCs) Included in the Kigali Amendment to the Montreal Protocol, Papanastasiou et al.

5

**Co-Editor Decision: Publish subject to technical corrections** (13 Apr 2018) by Andreas Engel
**Comments to the Author**: in general I am happy with most of the answers to the reviewers comments. However, inmany places, comments by the reviewers could have been accounted for by adding short clarifications in the text. This is particularly true for many comments by rev. #1. I suggest to add a few more additions to the text to react to
10    the reviewer comments. I consider this to be "technical corrections". Please revise the manuscript further based on the following reviewer comments which were acknowledged but no action was taken. This was especially true for the following comments:
      **Author Response**: We are glad that the manuscript has been accepted for publication in ACP. We have addressed the mostly tutorial "technical corrections" in a revised manuscript as requested by the editor and
15    described below.

**Editor Comment**: Abstract: I think it would be good to stress even more that what is derived here are estimtes and that these can be used as proxies in the absnce of experimantal data, in line with rev. #1 comments.
      **Author Response**: The manuscript was revised to include "estimates" in the title, which provides the basis for a strong emphasis that the manuscript provides estimated metrics. In this set of revisions, we have further revised
20    the abstract text as follows: "The results from this study provide estimated policy relevant GWP metrics for the HCFCs included in the Montreal Protocol in the absence of experimentally derived metrics.

**Editor Comment**: With respect to Rev. #1 comment on FRF (p.6. l. 37 of your response file): I agree that a short sentence stating what FRF describes would be beneficial. The same is true for ODP, GWP and GTP. This would enhance the readability of the manuscript.
25    **Author Response:** We have included tutorial statements in several locations in the manuscript. In this set of revisions, we have added the following (added reference given below):

*In the introduction, the first mention of GWP*: This necessitates knowledge of the global warming potentials (GWPs), a policy relevant metric representing the climate impact of a compound relative to $CO_2$, of all HCFCs involved in the baseline formulae.

30    *In the introduction, the first mention of ODP and GTP*: "The objective of the present work is to provide a comprehensive evaluation of the atmospheric lifetimes, ozone depletion potentials (ODPs), which represents the ozone depleting impact of a compound relative to a reference compound (see WMO (2014) and references within), GWPs, and global temperature change potentials (GTPs), another policy relevant metric representing the climate impact of a compound relative to $CO_2$, for the HCFCs listed in Annex C of the amended Protocol.

35    *In section 2.2, the first mention of FRF*: ".. *f* is the molecules fractional release factor (FRF), which denotes the fraction of the halocarbon injected into the stratosphere that has been dissociated (Solomon and Albritton, 1992), …".

Solomon, S., and D. L. Albritton, Time-dependent ozone depletion potentials for short- and long-term forecasts, Nature, 357, 33-37, 1992.

40    **Editor Comment**: With respect to Rev. #1 comment on band below 500 cm-1 (p.8, l. 5 of your response files): please include the blackbody emission of the earth in this argumentation as suggested by the reviewer.
      **Author Response:** We have gone to great lengths in our work to present the relationship between the molecules infrared absorption spectrum and the irradiance profile in the supporting information files. The SI files for each of the molecules included in this study show graphically the irradiance profile (taken from Hodnebrog et al.), the
45    molecules infrared absorption spectrum, and the calculated wavelength dependent radiative efficiency of the molecule. This clearly illustrates the relative importance of the <500 cm-1 region to the molecules radiative efficiency, which as we point out in the text is minor. To address this further we have added the following text in this section to the revised manuscript: "The contribution of vibrational bands in this region to the RE is quantified in our calculations and is usually minor, i.e., <1%. The Earth's irradiance profile, HCFC infrared

absorption spectra, and HCFC radiative efficiency spectra for each HCFC included in this study are included in the Supplementary Material (see Section 5)."

**Editor Comment**:  With respect to Rev. #1 comment on RE (p.10, l. 14 and 22 of your response files): please add the statement that this can only be done when measured accurate spectra are available.

    **Author Response:**  We have clarified the text by adding the following: "Note that molecules with strong absorption features near the large $CO_2$ and $O_3$ dips in the Earth's irradiance profile would have a greater sensitivity to shifts in the spectrum.  In such cases, direct laboratory studies would be invaluable in the determination of the molecules radiative properties."

**Editor Comment**:  With respect to the comment from rev.#2 about the use of FRF factors. the new FRF values presented in the papers by Leedham Elvidge and by Engel et al. are in most cases not very different from those used here. I would suggest to add a brief statements that a new method to calculate FRF values has been suggested by Ostermoeller et al. and applied by the two papers by Leedham Elvdige and Enge and that the general relationship between FRF and stratospheric lifetime is not laregely different (you might want to discuss the differences and include these as additional uncertainties).

    **Author Response:**  We agree with the Editor that the relationship between FRF and stratospheric lifetime developed in our work is not largely different from these recent publications.  In response to another Editor comment, we have inserted a description of what FRF means and cited the paper by Solomon and Albritton (1992).  In section 2.2 we have added the additional text which cites these recent papers as follows:  "The fractional release factor and global lifetime for CFC-11 were taken from the WMO (2014) ozone assessment report to be 0.47 and 52 years, respectively.  Note that a new method to calculate FRF has been suggested by Ostermoeller et al. (2017a,b), which has been applied by Leedham Elvdige et al. (2018) and Engel et al. (2018).  Overall, there is good agreement between the new method and the empirical parameterization applied in this work.".  The estimated uncertainties associated with our empirical parameterization have been addressed in our semi-empirical ODP estimate analysis.

New references:
Ostermoller, J., Bönisch, H., Jöckel, P., and Engel, A.: A new time-independent formulation of fractional release, Atmos. Chem. Phys., 17, 3785-3797, doi:10.5194/acp-17-3785-2017, 2017a.
Ostermoller, J., Bönisch, H., Jöckel, P., and Engel, A.: Corrigendum to "A new time-independent formulation of fractional release" published in Atmos. Chem. Phys., 17, 3785-3797, Atmos. Chem. Phys., 17, 3785-3797, doi:10.5194/acp-17-3785-2017-corrigendum, 2017b.
Leedham Elvidge, E., Bönisch, H., Brenninkmeijer, C. A. M., Engel, A., Fraser, P. J., Gallacher, E., Langenfelds, R., Mühle, J., Oram, D. E., Ray, E. A., Ridley, A. R., Röckmann, T., Sturges, W. T., Weiss, R. F., and Laube, J. C.: Evaluation of stratospheric age of air from $CF_4$, $C_2F_6$, $C_3F_8$, $CHF_3$, HFC-125, HFC-227ea and $SF_6$; implications for the calculations of halocarbon lifetimes, fractional release factors and ozone depletion potentials, Atmos. Chem. Phys., 18, 3369-3385, doi:10.5194/acp-18-3369-2018, 2018.
Engel, A., Bönisch, H., Ostermoller, J., Chipperfield, M., Dhomse, S., and Jöckel, P.: A refined method for calculating equivalent effective stratospheric chlorine, Atmos. Chem. Phys., 18, 601-619, doi:10.5194/acp-18-601-2018, 2018.

**Editor Comment**:  With respect to reviewer#3 comment to table 1 (p. 17, l. 9 in your author response), I suggest to mention briefly in the manuscript that different IR spectra are sometimes available and that a subjective choice has been made and give some reasoning on the choice.

    **Author Response:**  Our response (actually p. 18, l. 9).  As stated in our previous response, differences among various reported infrared absorption spectra are usually minor.  We have used data from our laboratory, the reliable measurements from Orkin et al., and the spectra reported in the highly reliable PNNL database whenever possible.  We have added the following text to the * footnote to Table 1: "Where multiple sources for infrared spectra are available, the spectra reported from the NOAA laboratory (McGillen et al., 2015) and the PNNL database (Sharpe et al., 2004) were used in the analysis."

**Editor Comment**:  With respect to reviewer#3 comment to p.10, l. 9-11 of the original manuscript: I suggest to include some reasoning for the way that the broadening has been calculated.

    **Author Response:**  The original reviewers comment was addressing whether including band broadening was an uncertainty or a sensitivity analysis.  The text below, taken from the manuscript clearly states that the motivation

for including broadening was to achieve better agreement with actual infrared absorption spectra. Although agreement is better, the functional form of the infrared band shape may not be Gaussian.

"The calculated spectra were broadened using a Gaussian broadening function with a FWHM (full width at half maximum) of 20 cm$^{-1}$, which reproduces the training dataset spectra reasonably well and provides a more realistic representation of the spectrum and overlap with Earth's irradiance profile. Note that the Gaussian broadening function may not necessarily be an accurate representation of the actual vibrational band shape."

5